# Efficient Conformal Prediction via Conformalized Meta-Learning from Noisy Labels

## Abstract

As a distribution-free uncertainty quantification method for machine learning models, conformal prediction constructs prediction sets with statistical coverage guarantees. However, in real deep-learning systems, the deep learners could be affected by training label noise, which leads to inefficiently large prediction sets. In this work, focusing on the classification task, we study and address such a robust learning issue within conformal prediction. We first empirically and theoretically analyze this problem. Then, to alleviate this issue, we propose an efficiency-aware conformalized meta-learning-based method, which directly minimizes the empirical size of prediction sets on meta data, aiming at rectifying the training loss. Experiments on datasets with both synthetic and real-world noise demonstrate that the proposed method can effectively enhance the efficiency of the prediction sets against training label noise.

## 1 Introduction

Nowadays, deep neural networks are widely used in various fields, such as healthcare and medicine (Ahmad et al., 2018; Berg et al., 2019). Despite their strong performance for single-point prediction, such models lack security guarantees. For example, the output of a single diagnosis is not always a good aid to the doctor's decision-making in medical diagnosis. It would be better for doctors to judge among a range of possible diagnoses relying on their experiences, which calls for uncertainty qualification (Abdar et al., 2021; Hüllermeier & Waegeman, 2021; Rudner & Toner, 2021). Such uncertainty can be classified by its source into two types: *epistemic* uncertainty, which can be reduced with more data or improved models, and *aleatory* uncertainty, which is inherent in data randomness (Fisch et al., 2022). Common uncertainty quantification methods include Bayesian modeling (Houlsby et al., 2011; Gal & Ghahramani, 2016; Kuleshov et al., 2018) and calibration techniques like Temperature Scaling (Guo et al., 2017) and Platt Scaling (Platt et al., 1999). Among them, *Conformal Prediction* (Vovk et al., 2005; Angelopoulos et al., 2023) stands out as a distribution-free framework that provides rigorous finite-sample coverage guarantees—predicting regions (e.g., intervals for regression and sets for classification) without requiring strong statistical assumptions. It requires only exchangeability (a weaker condition than i.i.d.) and is model-agnostic, making it directly applicable to any black-box machine learning model.

In multiclass classification tasks, valid, adaptive, and efficient prediction sets are highly desirable (Romano et al., 2020; Messoudi et al., 2020), which means that the obtained prediction sets should achieve desired marginal and conditional coverage, and also contain as few labels as possible. While all three aspects are important for conformal prediction, the efficiency is particularly of practical interest (Angelopoulos et al., 2021; Stutz et al., 2022; Bai et al., 2022; Dey et al., 2023; Sharma et al., 2023; Liu et al., 2024; Correia et al., 2024; Feldman & Romano, 2024b; Huang et al., 2024). In specific, provided the coverage properties are guaranteed, smaller prediction sets are generally more informative and help the decision-making process such as medical diagnosis, where such smaller prediction sets would enable doctors to make more accurate decisions and save time (Straitouri et al., 2023; Kapuria et al., 2024; Cresswell et al., 2024). Currently, two kinds of approaches have been developed to reduce the *inefficiency*, i.e., the size of prediction sets: designing novel *non-conformity* (NC) scores (Angelopoulos et al., 2021; Ding et al., 2023; Huang et al., 2024) and improving the underlying model, i.e., designing differentiable set size as part of the learning objective (Bellotti, 2021; Stutz et al., 2022; Liu et al., 2024). However, these studies generally assume that the training data

are noise-free, ignoring the training label noise, in which case the classifiers could be misleadingly trained.

In this work, we are trying to analyze and address the efficiency issue of the conformalized predictor under training label noise. Specifically, we first formally set up the problem of conformal prediction under training label noise. Then we provide empirical evidence and theoretical analysis to show how the training label noise could affect the prediction inefficiency. To enhance the efficiency of the conformalized predictor under training label noise, we propose an efficiency-aware meta-learning-based sample re-weighting method, such that the classifier can be robustly trained on the noisy training data guided by the efficiency-aware meta-objective, and thus is expected to produce smaller prediction sets in the test phase. Empirical evaluations on both synthetic and real noisy datasets demonstrate the effectiveness of the proposed method in reducing the inefficiency without sacrificing the coverage properties. To the best of our knowledge, this is the first attempt to directly explore the effect of training label noise on the inefficiency of prediction sets.

## 2 RELATED WORK

The label noise problem has been noted in recent advances of conformal prediction, particularly focusing on scenarios where noise contaminates the calibration set and consequently breaks the exchangeability assumption that is essential for marginal coverage guarantees (Penso & Goldberger, 2024; Cauchois et al., 2024; Einbinder et al., 2024; Sesia et al., 2024; Feldman & Romano, 2024b; Clarkson et al., 2024; Gong et al., 2025). For example, (Cauchois et al., 2024) introduced weak supervision conformal prediction, which preserves noise-consistent predictions by explicitly modeling the label corruption process in both calibration and test data. Then (Sesia et al., 2024) proposed coverage-robust calibration algorithms for clean test data through modified NC scores. More recently, (Einbinder et al., 2024) established theoretical bounds showing standard conformal methods remain valid but conservative with certain forms of label contamination. While existing approaches primarily address calibration-set noise, we study a distinct yet practically important scenario. In this setting, classifiers are trained on noisy data (reflecting real-world constraints) while maintaining exchangeability through a small, clean calibration set obtained via cost-effective manual verification (Ren et al., 2018)) at the test phase. This configuration inherently ensures marginal coverage, allowing us to focus on improving prediction efficiency. It should be mentioned that if the calibration set is also unfortunately contaminated with noise, existing methods (Sesia et al., 2024; Penso & Goldberger, 2024; Penso et al., 2025; Bortolotti et al., 2025) can be applied at the test phase to the predictors learned by our method, as illustrated in Section 6.1.

## 3 PRELIMINARIES

In this work, we consider multiclass classifiaction tasks. Let $\mathcal{X} \subset \mathbb{R}^d$ be the input space and $\mathcal{Y} = \{1, \ldots, K\}$ be the label space. Imagining that we have a model $f : \mathcal{X} \to \mathbb{R}^K$ parametered by $\mathbf{w}$, such as neural networks, and it approximately estimates the conditional probability $\mathbb{P}(Y = y | X = x)$. Then for each input $x \in \mathcal{X}$, we can use $f$ to predict the most likely label $\hat{y} = \arg\max_{y \in \mathcal{Y}} f(x)_y$, where $f(x)_y$ is the $y$-th component of $f(x; \mathbf{w})$. Similar to most previous studies (Angelopoulos et al., 2021; Stutz et al., 2022; Einbinder et al., 2022; Huang et al., 2023), we focus mainly on *Split Conformal Prediction* which can be applied end-to-end directly after model training (Vovk et al., 2005). This method starts by splitting the training data $\mathcal{D}_0$ into two disjoint subsets: $\mathcal{D}_0 = \mathcal{D}_{tr} \biguplus \mathcal{D}_{cal}$, where $\mathcal{D}_{tr} := \{(X_i^{tr}, Y_i^{tr})\}_{i=1}^N$ and $\mathcal{D}_{cal} := \{(X_i, Y_i)\}_{i=1}^n$. With such a splitting, we train the model only on $\mathcal{D}_{tr}$ while reserving a small number of i.i.d. data in $\mathcal{D}_{cal}$ unseen during model training for calibration. Following the convention of deep classifiers, we denote $f$ as the output of the softmax layer. Then we can compute a NC score $S_i = S(X_i, Y_i) \in \mathbb{R}$ for each calibration datum $(X_i, Y_i), i = 1, \ldots, n$. Generally, the NC score is high when the softmax value of the true label is low, i.e., when the model is badly wrong, and vice versa. After that, for a new test data $X_{n+1}$, its prediction set $\mathcal{C}_\alpha(X_{n+1}; \mathcal{D}_{cal})$ is constructed as follows:

$$\mathcal{C}_\alpha(X_{n+1}; \mathcal{D}_{cal}) := \{y \in \mathcal{Y} : S(X_{n+1}, y) \le \mathcal{Q}_{1-\alpha}(\{S_i\}_{i=1}^n)\}, \tag{1}$$

where $\mathcal{Q}_{1-\alpha}(\{S_i\}_{i=1}^n)$ denotes the $\lceil (1+n)(1-\alpha) \rceil / n$-th quantile of NC scores $\{S_i\}_{i=1}^n$, and $\alpha$ is the user-specified error rate. Then the constructed prediction sets provably achieve the $1 - \alpha$ marginal coverage, as formally guaranteed by the following theorem. In practical applications, the

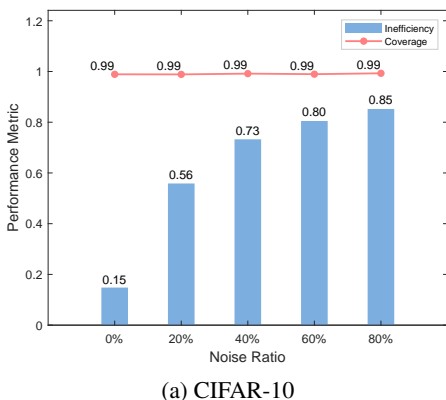 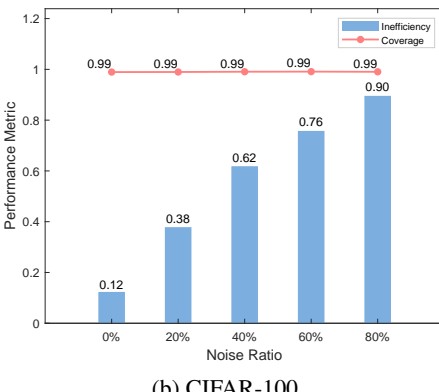

(a) CIFAR-10               (b) CIFAR-100

Figure 1: Empirical coverage and inefficiency of prediction sets by classifiers trained on datasets with training label noise for baseline (CE). The inefficiency is normalized by the number of all possible labels.

selection of NC score is extremely crucial. The two most classic methods are HPS (Sadinle et al., 2019), and APS (Romano et al., 2020). More details on NC scores can be found in Appendix A.1.

**Theorem 3.1** ((Vovk et al., 2005)). *Assume that examples* $(X_i, Y_i)$, $i = 1, \ldots, n+1$ *are exchangeable. For any NC score function, the prediction set* $\mathcal{C}_\alpha(X_{n+1}; \mathcal{D}_{cal})$ *is defined in Eq.* (1). *Then, the following holds:*

$$\mathbb{P}(Y_{n+1} \in \mathcal{C}_\alpha(X_{n+1}; \mathcal{D}_{cal})) \geq 1 - \alpha. \tag{2}$$

## 4 LEARNING CONFORMALIZED PREDICTORS FROM NOISY LABELS

### 4.1 PROBLEM SETUPS

In this work, we consider the situation that the training set is with label noise to train neural network models, while the calibration set is clean for constructing the prediction sets. In this setting, the marginal coverage guarantee in Eq. (2) still holds under the assumption of exchangeability between calibration and test data in Theorem 3.1. As mentioned in Section 1, this configuration allows us to focus on the effect of training label noise on prediction efficiency. Note that in practice, the calibration set may also contain label noise. Nevertheless, in such situations, we can manually refine the labels at an affordable cost since the calibration set is generally small. In addition, previously developed methods dealing with calibration label noise (Sesia et al., 2024; Penso & Goldberger, 2024; Penso et al., 2025; Bortolotti et al., 2025) can also be applied at the test phase, as illustrated in Section 6.1. In the following part, we first empirically show the effect of training label noise on the size of the prediction set, and then theoretically analyze it.

### 4.2 EMPIRICAL OBSERVATION

We conduct a series of experiments to empirically study the effect of training label noise on the prediction set's size. Specifically, we train deep classifiers on noisy training sets and construct the prediction sets using HPS (Sadinle et al., 2019) for test points with clean calibration sets, and then observe the average set sizes. For CIFAR-10 and CIFAR-100 datasets, we manually add symmetric noise with different ratios (see Section 6.1 for its generation). The empirical results are shown in Figure 1.

We can see that, in all situations, the marginal coverage can be guaranteed, since the exchangeability assumption between calibration and test data still holds. However, the average size of the prediction sets, i.e., *inefficiency*, becomes larger as the noise ratio increases. For example, under a noise ratio of 80%, the prediction set contains over 85% of all candidate labels. These results could be intuitively explained by Figure 2: on the one hand, the classifier trained on the noisy training data tends to make poor predictions on the calibration set, and thus the distribution of NC scores can largely deviates from that of the training noise-free situation; on the other hand, such a classifier also tends to make

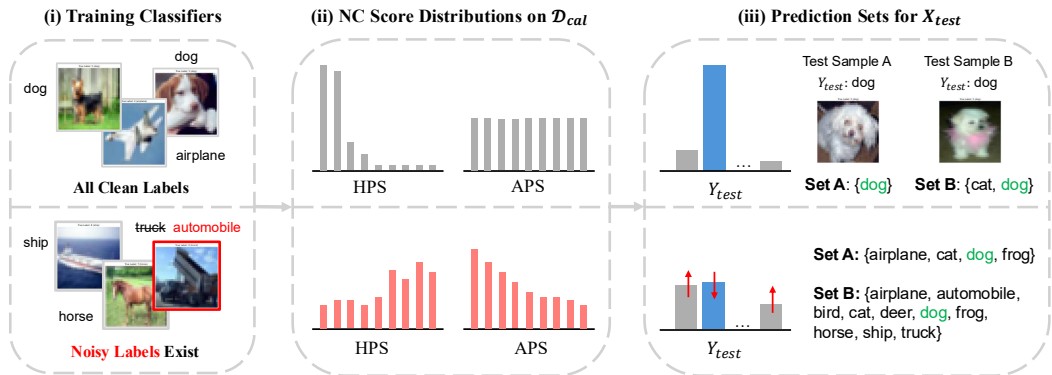

Figure 2: Intuitive illustration for the effects of label noise on split conformal prediction. **Left:** Training images with all clean labels v.s. presence of noisy labels. **Middle:** The empirical distributions of NC scores on the calibration set. When training label noise exists, the scores by HPS tend to take higher values, and the distributions of scores by APS tend to be non-uniform. **Right:** Prediction sets for test data. Under the same $1 - \alpha$ marginal coverage, the prediction sets become larger by the classifier trained with label noise due to its confused prediction.

confused prediction on the test points, which, together with the deviated NC score distribution, results in large prediction sets to achieve desired coverage.

## 4.3 THEORETICAL ANALYSIS

Now we try to anlayze the previous observations with mathematical tools. Following the previous studies (Dhillon et al., 2024; Zecchin et al., 2024), we analyze the expected prediction set size $\mathbb{E}\big[\big|\mathcal{C}_\alpha(X_{n+1}; \mathcal{D}_{cal})\big|\big]$ for the finite sample, where $|\cdot|$ denotes the counting measure for the discrete label space $\mathcal{Y}$ in classification tasks. For any NC score function $S(X, Y)$ and $s \in \mathcal{R}$, $F_{cal}(s)$ is defined as $F_{cal}(s) = \mathbb{P}[S(X, Y) < s]$, where the probability is computed over the calibration data $\mathcal{D}_{cal} \sim \mathcal{P}_{(X,Y)}$. And we define $n_\alpha = \lceil (1 - \alpha)(n + 1) \rceil - 1$, where $\alpha$ is the user-specified error rate. Then the expected prediction set size satisfies Theorem B.1 (Dhillon et al., 2024) in Appendix B.1. For convenience, we denote $\hat{Y}$ is the most likely predicted label by the classifier $f$ for the input $X$, while $Y$ is the true label. When the NC score is simply computed using the 0-1 loss, i.e., $S(X, Y) = \mathbb{I}\{\hat{Y} \neq Y\}$, we can derive the following conclusions (whose proof is given in Appendices B.2 and B.3):

**Theorem 4.1** (**0-1 NC score**). *Under the assumptions of Theorem B.1, if the NC score is 0-1 loss defined as $S(X, Y) = \mathbb{I}\{\hat{Y} \neq Y\}$, the expected size of the split conformal prediction sets satisfies*

$$\mathbb{E}\big[\big|\mathcal{C}_\alpha(X_{n+1}; \mathcal{D}_{cal})\big|\big] = 1 + \mathbb{P}\big[B\big(n, p_{cal}\big) \leq n_\alpha\big]\big(|\mathcal{Y}| - 1\big), \quad (3)$$

*where $p_{cal} \triangleq \mathbb{P}_{cal}\big[\hat{Y} = Y\big]$ denotes the test accuracy, and $B(n, \pi)$ denotes a binomial random variable with $n$ trials and success probability $\pi$.*

Theorem 4.1 explicitly associates the classification accuracy on the calibration data with the expected set size. Specifically, the higher accuracy lowers the probability $\mathbb{P}\big[B\big(n, p_{cal}\big) \leq n_\alpha\big]$ and thus the expected size $\mathbb{E}\big[\big|\mathcal{C}_\alpha(X_{n+1}; \mathcal{D}_{cal})\big|\big]$, and vice versa. Consider two extreme cases, in which the classifier is perfect with 100% accuracy, and the classifier purely randomly guesses the label, we can get the expected sizes 1 and $|\mathcal{Y}|$, respectively. In addition, as suggested by Proposition 4.2 (Appendix B.3 for asymmetric noise), if we further assume that the Bayes optimal classifier can be obtained, then the expected size of the predictions can solely depend on the noise ratio of the training data.

**Proposition 4.2.** *Assuming that the classifier is Bayes optimal on training set $\mathcal{D}_{tr}$, and the clean posterior satisfies $P[Y = k|X = x] > P[Y = i|X = x](i \neq k)$, for the true label $k$ of $X$. If $\mathcal{D}_{tr}$*

*contains symmetric noise defined in Appendix A.2, then*

$$p_{cal} \approx \sum_{k=1}^{K} \frac{\pi_k m_k(X)}{1 + \sum_{i \neq k} b^{\left(m_k(x) - m_i(x)\right)\left(\frac{K}{K-1}\epsilon - 1\right)}}, \tag{4}$$

*where $\pi_k \triangleq \mathbb{P}_{cal}[Y = k]$, $m_k(x) \triangleq \mathbb{P}_{cal}[Y = k | X = x]$. The accuracy of the classifier on the clean test data $p_{cal}$ decreases as the noise ratio $\epsilon$ increases.*

As suggested by Dhillon et al. (2024) and Zecchin et al. (2024), unlike 0-1 loss, the expected set size for other NC scores couldn't be derived without additional assumptions. So when analyzing other NC scores, such as HPS and APS, we turn to focus on their empirical distributions under training noise. We find that as the noise ratio increases, HPS scores become larger and APS scores become non-uniform in Figure 3 (in Appendix B.4). It could be explained that a higher noise ratio reduces the conformity between training and calibration data, which indeed leads to larger prediction sets. More discussions of score distribution are given in Appendix B.4. Furthermore, Stutz et al. (2022) also found that the accuracy didn't directly control the set size in general experiments.

# 5 EFFICIENCY-AWARE CONFORMALIZED META-WEIGHT-NET

It suggests that existing traditional robust learning methods (Song et al., 2022), such as robust learning function or loss adjustment, try to improve accuracy; however, they don't focus on prediction uncertainty. As a post-hoc framework, CP can be directly combined with these robust strategies, but our experimental results in Section 6 show that the set size of these methods, such as Meta-Weight-Net (CE-MWN) (Shu et al., 2019), could be rather large. The main issue lies in the decoupling of the training stage and the CP calibration. This motivates us to develop a new method to address this problem using a meta-learning strategy.

## 5.1 LEARNING OBJECTIVE

Recently, the meta-learning (Finn et al., 2017; Hospedales et al., 2022; Vettoruzzo et al., 2024) methodology has been applied to deal with the robust learning problem and has shown its effectiveness (Ren et al., 2018; Shu et al., 2019; Wu et al., 2021; Wang et al., 2020; Tu et al., 2023). The main idea of meta-learning-based robust learning is first to maintain a small set of clean meta data, which can be obtained by manual label refinement (Ren et al., 2018), to simulate the test distribution, and then to use a certain loss evaluated on the meta data to specify the mechanism for rectifying the training loss, such as example re-weighting (Ren et al., 2018; Shu et al., 2019), label correction (Wu et al., 2021; Tu et al., 2023) and noise transition estimation (Wang et al., 2020). Following this methodology, we propose to use a small set of clean meta data to simulate the process of split conformal prediction and construct an efficiency-aware meta loss to guide the classifier training on the noisy training data. Denoting $\mathcal{D}_{tr} := \{(x_i^{tr}, y_i^{tr})\}_{i=1}^{N}$ as noisy training data and $\mathcal{D}_{meta} := \{x_j^{meta}, x_j^{meta}\}_{j=1}^{M}$ as the clean meta data, the overall learning objective can be formulated as the following bi-level optimization:

$$\Theta^* = \arg\min_{\Theta} \frac{1}{M} \sum_{j=1}^{M} \left| \mathcal{C}_\alpha(x_j^{meta}; \mathbf{w}^*(\Theta)) \right|, \quad s.t. \; \mathbf{w}^*(\Theta) = \arg\min_{\mathbf{w}} \frac{1}{N} \sum_{i=1}^{N} \mathcal{V}\left(L_i^{tr}(\mathbf{w}); \Theta\right) L_i^{tr}(\mathbf{w}), \tag{5}$$

where $L_i^{tr}(\mathbf{w}) = \ell\left(y_i^{tr}, f(x_i^{tr}; \mathbf{w})\right)$ for simplicity, and $\mathcal{V}(\cdot; \Theta)$ is the weighting function parameterized by a multilayer perception (MLP), whose parameters are $\Theta$.

Before presenting the algorithm for solving the above model, we first discuss more about it. Compared with Meta-Weight-Net (MWN) (Shu et al., 2019), we adopt the same example re-weighting strategy to rectify the training loss, but use the empirical set size as the meta loss instead of the cross-entropy (CE) loss, which better fits our goal of improving the efficiency. Specifically, the CE loss mainly focuses on the accuracy of the predictor. Therefore, directly minimizing the set size in the meta objective would be more reasonable, and its advantages over the CE loss will be empirically illustrated in Section 6. As for training loss rectification, other mechanisms such as label correction (Wu et al., 2021) and noise transition estimation (Wang et al., 2020) may also be applied and should be worth exploring in the future. Nevertheless, the simple example re-weighting strategy has shown effectiveness in our empirical studies.

## 5.2 OBJECTIVE RELAXATION

The meta loss in Eq. (5) can be explicitly written as

$$\big|\mathcal{C}_\alpha(x_j^{meta}; \mathbf{w}^*(\Theta))\big| = \sum_{y \in \mathcal{Y}} \mathbb{I}\{\mathcal{Q}_{1-\alpha}(\{S_i\}_{i=1}^n) \geq S(x_j^{meta}, y)\}, \quad (6)$$

which is non-differentiable due to the discrete value it takes and the sorting operation in the quantile computation. Therefore, we first need to relax it to a differentiable function.

Inspired by ConfTr (Stutz et al., 2022) and several studies thereafter (Huang et al., 2023; Yan et al., 2024), we split meta data into two subsets: $\widetilde{\mathcal{D}}_{cal} = \{(x_{j_1}, y_{j_1})\}_{j_1=1}^{M/2}$ and $\widetilde{\mathcal{D}}_{pred} = \{(x_{j_2}, y_{j_2})\}_{j_2=1}^{M/2}$. $\widetilde{\mathcal{D}}_{cal}$ is first leveraged to compute the quantile $\widetilde{\mathcal{Q}}_{1-\alpha}(\mathbf{w}(\Theta))$ through differentiable sorting networks (Cuturi et al., 2019; Blondel et al., 2020), and then $\widetilde{\mathcal{D}}_{pred}$ is used to compute the smoothed set size as our meta loss. Specially, the loss for the $j_2$-th datum can be relaxed to $\Omega_{\alpha,j_2}(\mathbf{w}(\Theta)) \triangleq \sum_{y \in \mathcal{Y}} \sigma\left(\frac{\widetilde{\mathcal{Q}}_{1-\alpha}(\mathbf{w}(\Theta)) - S(x_{j_2}, y)}{T}\right)$, where $\sigma(\cdot)$ is the sigmoid function defined as $\sigma(z) = 1/(1 + \exp(-z))$, and $T$ is the temperature hyperparameter. If $T \to 0$, $\Omega_{\alpha,j_2}(\mathbf{w}(\Theta))$ will approach $\big|\mathcal{C}_\alpha(x_{j_2}; \mathbf{w}(\Theta))\big|$. Using such a differentiable relaxation, the learning goal becomes the following bi-level optimization:

$$\Theta^* = \arg\min_\Theta \frac{1}{M/2} \sum_{j_2=1}^{M/2} \Omega_{\alpha,j_2}(\mathbf{w}^*(\Theta)), \quad s.t. \ \mathbf{w}^*(\Theta) = \arg\min_\mathbf{w} \frac{1}{N} \sum_{i=1}^N \mathcal{V}(L_i^{tr}(\mathbf{w}); \Theta) L_i^{tr}(\mathbf{w}). \quad (7)$$

## 5.3 LEARNING PROCESS

The bi-level optimization presented in the previous subsection is still difficult to solve due to its nested structure. There are various studies devote to solving such a bi-level optimization (Liu et al., 2022), while we adopt the one-step gradient approximation strategy presented in MAML (Finn et al., 2017), which has shown its simplicity and effectiveness in the meta-learning-based robust learning methods (Ren et al., 2018; Shu et al., 2019; Wang et al., 2020).

Specially, in each iteration during the training process, a mini-batch of training data $\{(x_i^{tr}, y_i^{tr})\}_{i=1}^{n_0}$ and a mini-batch of meta data $\{(x_j^{meta}, y_j^{meta})\}_{j=1}^{m_0}$ are sampled. Then we use the one-step gradient update (Finn et al., 2017) to approximate the lower-level optimization:

$$\hat{\mathbf{w}}^{(t)}(\Theta) = \mathbf{w}^{(t)} - \beta_1 \times \frac{1}{n_0} \sum_{i=1}^{n_0} \mathcal{V}(L_i^{tr}(\mathbf{w}^{(t)}); \Theta) \nabla_\mathbf{w} L_i^{tr}(\mathbf{w})\big|_{\mathbf{w}^{(t)}}. \quad (8)$$

Next, we update $\Theta$ with one-step gradient descent:

$$\Theta^{(t+1)} = \Theta^{(t)} - \beta_2 \times \frac{1}{m_0/2} \sum_{j_2=1}^{m_0/2} \nabla_\Theta \Omega_{\alpha,j_2}(\hat{\mathbf{w}}^{(t)}(\Theta))\big|_{\Theta^{(t)}}, \quad (9)$$

where $\hat{\mathbf{w}}^{(t)}(\Theta)$ is used to approximate $\mathbf{w}^*(\Theta)$. After updating $\Theta$, the current weight function $\mathcal{V}(L_i^{tr}(\mathbf{w}^{(t)}); \Theta^{(t+1)})$ is used to update the parameter of the classifier $\mathbf{w}$:

$$\mathbf{w}^{(t+1)} = \mathbf{w}^{(t)} - \beta_1 \times \frac{1}{n_0} \sum_{i=1}^{n_0} \mathcal{V}(L_i^{tr}(\mathbf{w}^{(t)}); \Theta^{(t+1)}) \nabla_\mathbf{w} L_i^{tr}(\mathbf{w})\big|_{\mathbf{w}^{(t)}}. \quad (10)$$

The whole learning process referred to as Conformalized Meta-Weight-Net (Conf-MWN), is summarized in Algorithm 1 .

## 5.4 THEORETICAL PROPERTIES

Inspired by Zhao et al. (2019), we provide the generalization result in Theorem 5.1 (proof is given in Appendix C.1). It indicates that the proposed meta-learning model approaches the optimal weight at a rate $\mathcal{O}(\sqrt{d \ln(M)/M})$.

**Theorem 5.1.** *Assume that meta loss $\Omega(X^{meta}; \hat{\mathbf{w}}(\Theta))$ is $\lambda$-Lipschitz continuous with respect to $\Theta$. Let $\Theta \in \mathbb{B}^d$ be the parameter of training weighting function in a $d$-dimensional unit ball, and $M$ be the number of meta data. If $(X^{meta}, Y^{meta}) \sim \mathcal{P}^{cal}$, then define the generalization risk as:*

$$R(\hat{\mathbf{w}}(\Theta)) = \mathbb{E}[\Omega(X^{meta}; \hat{\mathbf{w}}(\Theta))]. \quad (11)$$

---

**Algorithm 1** The Conf-MWN Learning Algorithm

---

**Input:** Training data set $\mathcal{D}_{tr}$, meta data set $\mathcal{D}_{meta}$, batch size $n_0$, $m_0$, and max iteration $T$.
**Output:** The weight-net's parameters $\Theta^{(T)}$, and the classifier's parameters $\mathbf{w}^{(T)}$.
 1: **for** $t = 0$ **to** $T - 1$ **do**
 2:     $\{x_i^{tr}, y_i^{tr}\}_{i=1}^{n_0} \leftarrow$ Sample mini-batch $(\mathcal{D}_{tr}, n_0)$ *# Training data with noisy labels.*
 3:     $\{x_j^{meta}, y_j^{meta}\}_{j=1}^{m_0} \leftarrow$ Sample mini-batch $(\mathcal{D}_{meta}, m_0)$ *# Meta data with clean labels.*
 4:     Formulate the classifier parameter $\hat{\mathbf{w}}^{(t)}(\Theta)$ by Eq. (8).
 5:     Randomly split $\{x_j^{meta}, y_j^{meta}\}_{j=1}^{m_0}$ in half.
 6:     For $\widetilde{\mathcal{D}}_{cal}$: update differentiable quantile $\widetilde{\mathcal{Q}}_{1-\alpha}(\hat{\mathbf{w}}^{(t)}(\Theta))$.
 7:     For $\widetilde{\mathcal{D}}_{pred}$: compute $\Omega_{\alpha, j_2}(\hat{\mathbf{w}}^{(t)}(\Theta))$.
 8:     Update $\Theta^{(t+1)}$ by Eq. (9).
 9:     Update $\mathbf{w}^{(t+1)}$ by Eq. (10).
10: **end for**

---

*Let $\Theta^* = \arg\min_{\Theta \in \mathbb{B}^d} R(\hat{\mathbf{w}}(\Theta))$ be the optimal parameter in the unit ball, and $\hat{\Theta} = \arg\min_{\Theta \in \mathcal{A}} \hat{R}(\hat{\mathbf{w}}(\Theta))$ be the empirically optimal among a candidate set $\mathcal{A}$. With probability at least $1 - \delta$, we have*

$$R(\hat{\mathbf{w}}(\hat{\Theta})) - R(\hat{\mathbf{w}}(\Theta^*)) \leq \frac{3\lambda + \sqrt{4d\ln(M) + 8\ln(2/\delta)}}{\sqrt{M}}. \tag{12}$$

In addition, we can show the theoretical convergence of the learning process (see Appendix C.2).

## 6 EXPERIMENTS

To verify the effectiveness of Conf-MWN, we conduct experiments on datasets with both synthetic training label noise and real-world noise. We mainly compare our method with direct training using CE on noisy data, while also considering the CE-based MWN (Shu et al., 2019), denoted as CE-MWN, as a reference. Three metrics are used for evaluation, which are empirical coverage (Cov.), inefficiency (Ineff.) and accuracy (Acc.), defined as Cov. $:= \frac{1}{|\mathcal{D}_{test}|} \sum_{X_i \in \mathcal{D}_{test}} \mathbb{I}[Y_i \in \mathcal{C}_\alpha(X_i; \mathcal{D}_{cal})]$, Ineff. $:= \frac{1}{|\mathcal{D}_{test}|} \sum_{X_i \in \mathcal{D}_{test}} |\mathcal{C}_\alpha(X_i; \mathcal{D}_{cal})|$ and Acc. $:= \frac{1}{|\mathcal{D}_{test}|} \sum_{X_i \in \mathcal{D}_{test}} \mathbb{I}[\hat{Y}_i = Y_i]$, respectively. Among them, it always holds that Cov. $\approx 1 - \alpha$ due to the marginal coverage guarantee (Angelopoulos et al., 2023), and thus we focus mainly on Ineff., i.e., the average size of the prediction sets. Acc. is also reported as a reference since it is correlated to Ineff.. We mainly focus on HPS with $\alpha = 0.01$ in the main text, and put the extended results, including more error rates and other scores, in the Appendices E.1 and E.2.

### 6.1 DATASETS WITH SYNTHETIC LABEL NOISE

We adopt two common datasets CIFAR-10 and CIFAR-100 (Krizhevsky, 2009), and consider two settings of label noise (Song et al., 2022): (1) **Symmetric Noise.** The true label is corrupted by a noise transition matrix $T \in [0,1]^{K \times K}$, whose element $T_{ij} := p(Y' = j | Y = i)$ is the probability of the true label $i$ being flipped into a corrupted label $j$ with equal probability. For a noise rate $\tau \in [0,1]$, $T_{ij} = 1 - \tau$ for $j = i$ and $T_{ij} = \frac{\tau}{K-1}$ for any $j \neq i$. (2) **Asymmetric Noise.** The true label is more likely to be mislabeled into a particular label, and thus $T_{ij} = 1 - \tau$ for $j = i$, while $T_{ij} = \tau$ for a specific $j \neq i$. We randomly split 1000 images with clean labels from the training set as meta-data $\mathcal{D}_{meta}$ and calibration data $\mathcal{D}_{cal}$ respectively, and add label noise to the rest as the noisy training data $\mathcal{D}_{tr}$. More detailed experimental settings are provided in Appendix D.1.

**Decreasing of the prediction set's size with Conf-MWN.** Tables 1 and 2 show the average performance of different methods on CIFAR-10 and CIFAR-100 datasets, respectively, with symmetric (sym.) noise (results with asymmetric noise are shown in Tables 9 and 10 in Appendix E.3). It can be observed that our Conf-MWN outperforms CE in Ineff. under almost all noise types and

Table 1: Results on CIFAR-10 datasets with symmetric (sym.) noise. The performance is averaged over 3 random runs and the best is highlighted in **Bold**.

| Methods | No Noise | | | Sym.-20% | | | Sym.-40% | | |
|---|---|---|---|---|---|---|---|---|---|
| | Cov.% | Ineff. | Acc.% | Cov.% | Ineff. | Acc.% | Cov.% | Ineff. | Acc.% |
| CE | 98.81 | 1.51 | 92.01 | 98.95 | 5.31 | 87.55 | 99.03 | 7.42 | 83.78 |
| CE-MWN | 98.89 | 1.50 | 92.29 | 98.98 | 2.66 | 89.78 | 98.81 | 3.03 | 86.62 |
| Conf-MWN | 99.03 | 1.50 | 92.68 | 98.82 | **1.98** | 90.18 | 98.99 | **2.85** | 87.08 |

Table 2: Results on CIFAR-100 datasets with symmetric (sym.) noise. The performance is averaged over 3 random runs and the best is highlighted in **Bold**.

| Methods | No Noise | | | Sym.-20% | | | Sym.-40% | | |
|---|---|---|---|---|---|---|---|---|---|
| | Cov.% | Ineff. | Acc.% | Cov.% | Ineff. | Acc.% | Cov.% | Ineff. | Acc.% |
| CE | 98.87 | 13.20 | 69.18 | 99.01 | 40.69 | 62.23 | 98.96 | 58.67 | 55.71 |
| CE-MWN | 98.87 | 12.08 | 69.79 | 98.94 | 27.15 | 64.68 | 98.86 | 42.09 | 58.32 |
| Conf-MWN | 98.95 | 12.57 | 69.73 | 99.03 | **24.50** | 65.44 | 98.91 | **33.60** | 60.20 |

all noise ratios. For example, under 20% symmetric noise on CIFAR-10, the average prediction sets' size by Conf-MWN is only 1.98, which achieves over 60% reduction compared with that of CE, greatly improving the efficiency and informativeness of prediction sets. Besides, we also observe that while CE-MWN already performs well, our method can further consistently improve the `Ineff.` metric, since it directly minimizes the efficiency-aware meta loss. The convergence curves of inner and outer loss are shown in Figure 4 (Appendix C.3). More analysis of hyperparameters such as different classifiers, the number of meta and calibration data, the temperature $T$ can be found in Appendix F.

**The improvement in the distribution of HPS and APS scores.** In addition to these quantitative results, we draw the empirical *probability density function* (PDF) of the NC scores (HPS and APS) on the calibration set by Conf-MWN across different noise ratios in Appendix E.4 (Figure 5). The figure empirically suggests the mechanism of the effectiveness of Conf-MWN, that it can promisingly correct the distribution of the NC scores, though the classifier is trained on noisy data.

**Comparison with robust learning or conformal training methods.** More experimental comparisons with classic robust learning methods such as GCE (Zhang & Sabuncu, 2018) and Co-teaching (Han et al., 2018) are provided in Appendix E.5 (Tables 11 and 12). We also find that when CIFAR-100 training set contains 40% symmetric noise, GCE has an accuracy rate 13.46% higher than Co-teaching, but the set size is larger. This highlights that traditional robust learning methods aimed at improving accuracy may not be fully adapted to improving the efficiency of the prediction set. We also conduct experiments with only ConfTr at 99% coverage and show the results in Appendix E.6 (Tables 13). The results show that, without the robust consideration, ConfTr tends to be significantly affected by training label noise.

**Both training and calibration data contain label noise.** In practice, we also face the situation that the calibration set contains label noise. In this case, we can still apply Conf-MWN to train the classifier, and then use methods dealing with calibration noise (Cauchois et al., 2024; Clarkson et al., 2024; Einbinder et al., 2024; Feldman & Romano, 2024a; Sesia et al., 2024; Penso & Goldberger, 2024; Penso et al., 2025) to do split conformal prediction, instead of the standard way. Here, we show some empirical results by using the NACP (Penso et al., 2025) method in Table 3, and more results are provided in Appendix E.7 (Tables 14, 15 and 16). It can be seen that, due to the violation of exchangeability between the calibration and test sets, the coverage is no longer guaranteed with the standard conformal prediction. Nevertheless, with NACP, the coverage can be ameliorated toward the target, and Conf-MWN can still significantly enhance the efficiency of the prediction sets.

**The adaptiveness of the prediction sets using Conf-MWN.** We also show examples of prediction sets under training with CE and our method in Appendix G (Figures. 9 and 10). As can be seen, our method effectively excludes the irrelevant labels in the prediction sets. Besides, some interesting observations can be drawn, for example, from Figure 9: (1) Our method exactly reflects the ambiguity

Table 3: Results on CIFAR-100 datasets with both training and calibration sets containing symmetric (sym.) label noise. The best performance is highlighted in **Bold**.

| | Sym.-20% | | | | Sym.-40% | | | |
| | Cov.% | | Ineff. | | Cov.% | | Ineff. | |
| Scores | CE | Conf-MWN | CE | Conf-MWN | CE | Conf-MWN | CE | Conf-MWN |
|---|---|---|---|---|---|---|---|---|
| HPS | 99.95 | 99.97 | 9.48 | 9.19 | 99.92 | 99.95 | 9.68 | 9.58 |
| HPS+NACP | 99.21 | 99.35 | 6.85 | **2.56** | 99.33 | 99.26 | 8.20 | **5.40** |

between deer and horse for the third example; (2) Comparing the first and second ones, the ears of the dog in the second are shorter, and thus can reasonably be confused with a cat. These examples suggest that our method not only provides more informative prediction sets but also more faithfully reflects the prediction uncertainty for a test point. As conditional coverage is also an important aspect of conformal prediction, we have also empirically studied it, and the corresponding results are provided in Appendix E.8. It can be observed that Conf-MWN obtains at least comparable results against the CE baseline, showing that it can maintain a reasonable conditional coverage in general. The above results suggest that our method can produce prediction sets with desired adaptiveness.

## 6.2 DATASETS WITH REAL-WORLD LABEL NOISE

We consider three real-world noisy datasets: CIFAR-10N, CIFAR-100N (Wei et al., 2022), and Food-101N (Lee et al., 2018). CIFAR-10N and CIFAR-100N are respectively generated from the training datasets of CIFAR-10 and CIFAR-100 with human-annotated real-world noisy labels collected from Amazon Mechanical Turk. For CIFAR-10N datasets, there are three types of noise, which are *aggregate*, *random* and *worst*, and the noise ratio of the three types are 9.03%, $\approx 18\%$ and 40.21%, respectively. The details of the adopted models and training settings are provided in Appendix D.2.

We report partial results on CIFAR-10N datasets in Table 4, and put more results in Appendix E.9 (Tables 23, 24 and 25) due to page limitation. Similar to the previous analysis, Conf-MWN performs better than CE and CE-MWN in `Ineff.` under all scenarios. Besides, compared with the CE baseline, the reduction in `Ineff.` is also very significant. All the above results demonstrate the robustness and efficiency of the proposed Conf-MWN against training label noise.

Table 4: Results on CIFAR-10N datasets with real noise. The performance is averaged over 3 random runs and the best is highlighted in **Bold**.

| | Aggressive-9.03% | | | Random 1-17.23% | | | Worst-40.21% | | |
| Method | Cov.% | Ineff. | Acc.% | Cov.% | Ineff. | Acc.% | Cov.% | Ineff. | Acc.% |
|---|---|---|---|---|---|---|---|---|---|
| CE | 98.89 | 3.63 | 89.06 | 99.09 | 4.28 | 87.26 | 98.94 | 5.70 | 80.24 |
| CE-MWN | 99.07 | 2.71 | 90.09 | 98.93 | 2.68 | 89.07 | 99.03 | 3.64 | 83.19 |
| Conf-MWN | 98.94 | **2.19** | 90.66 | 98.95 | **2.55** | 89.20 | 98.95 | **3.53** | 82.67 |

## 7 CONCLUSION

In this paper, we have studied the problem of conformal prediction under training label noise. We have first empirically shown that the training label noise can make the prediction sets less efficient, and then provided explanations with mathematical tools. To alleviate such an efficiency issue, we have proposed an efficiency-aware conformalized meta-learning method, Conf-MWN, which directly minimizes the empirical prediction set's size on the meta data to guide re-weighting the training loss against label noise. Experiments on multiple diverse datasets have demonstrated the effectiveness of Conf-MWN in enhancing the efficiency of the prediction sets under training label noise. In the future, we will explore more training rectification mechanisms within the meta-learning framework.

## REPRODUCIBILITY STATEMENT

We have provided the experimental designs and details in Section 6 and Appendix D. The theoretical results can be found in Sections 4.3 and 5.4, and more analysis are in Appendices B and C. The real datasets used in the experiments is publicly accessible, and we have provided a detailed description in Sections 6.1 and 6.2. Although the code is not included in this submission, the specific experimental details are explained in Appendix D, and the code will be publicly available if the paper is accepted.

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

## A    MORE DETAILS

### A.1    THE NC SCORES

To obtain prediction sets with minimal average size under exact $1-\alpha$ marginal coverage guarantees, HPS (Homogeneous Prediction Set) (Sadinle et al., 2019) uses the NC score

$$S(X_i, Y_i) = 1 - f(X_i)_{Y_i}, \tag{13}$$

and penalizes low-probability predictions of the true label. For achieving conditional coverage, APS (Adaptive Prediction Set) (Romano et al., 2020) implements a distinct strategy via the randomized NC score:

$$S(X_i, Y_i) = \sum_{y \in \mathcal{Y}} f(X_i)_y \cdot \mathbb{I}\{f(X_i)_y > f(X_i)_{Y_i}\} + f(X_i)_{Y_i} \cdot u, \tag{14}$$

where the auxiliary randomness $u \sim \text{Uniform}(0,1)$ enables exact finite-sample coverage. To further reduce the size of the prediction set and thereby enhance the efficiency, RAPS (Regularized Adaptive Prediction Set) (Angelopoulos et al., 2021) and SAPS (Sorted Adaptive Prediction Set) (Huang et al., 2024) based on APS were developed. Furthermore, the conditional coverage $\mathbb{P}(Y_{n+1} \in \mathcal{C}_\alpha(X_{n+1}; \mathcal{D}_{cal})|X_{n+1}) \geq 1 - \alpha$ is satisfied if and only if $S(X_i, Y_i)$ in Eq. (14) follows a uniform distribution for each $X_i$ (Romano et al., 2020; Einbinder et al., 2022).

### A.2    THE LABEL NOISE ASSUMPTIONS FOR PROPOSITIONS 4.2

Similar to previous label noise studies (Shu et al., 2019; Oyen et al., 2022), we introduce the noise assumptions for the training label made in Propositions 4.2. Given the noise ratio $\epsilon$ ($0 \leq \epsilon \leq 1$), the distribution of symmetric noisy labels is defined as

$$\eta_{ki} = \begin{cases} 1 - \epsilon, & \text{for } i = k \\ \frac{\epsilon}{K-1}, & \forall i \in \{1, \dots, K\} \backslash k. \end{cases} \tag{15}$$

The asymmetric noisy label distribution is defined as

$$\eta_{ki} = \begin{cases} 1 - \epsilon, & \textit{for } i = k \\ \epsilon \cdot t_{ki}, & \forall i \in \{1, \dots, K\} \setminus k \textit{ and } \sum_{i \neq k} |t_{ki}|^0 = s \textit{ and } \sum_{i \neq k} t_{ki} = 1, \end{cases} \tag{16}$$

where $s(1 \leq s \leq K-1)$ denotes the spread of the noisy label distribution.

## B    THEORETICAL ANALYSIS OF THE EXPECTED SET SIZE

### B.1    THE EXPECTED SIZE OF PREDICTION SETS

**Theorem B.1** ((Dhillon et al., 2024)). *If the test and the calibration NC scores are independent of each other, and the calibration NC scores are i.i.d., the expected size of the split conformal prediction sets satisfies*

$$\mathbb{E}\big[|\mathcal{C}_\alpha(X_{n+1}; \mathcal{D}_{cal})|\big] = \sum_{y \in \mathcal{Y}} \mathbb{E}\left\{ \mathbb{P}\left[ B\left(n, F_{cal}\big(S(X_{n+1}, y)\big)\right) \leq n_\alpha \right] \right\}, \tag{17}$$

*where $B(n, \pi)$ denotes a binomial random variable with $n$ trials and success probability $\pi$.*

### B.2    PROOF OF THEOREM 4.1

*Proof.* For any specified error rate $\alpha$, the expected set size, according to Theorem B.1, is

$$\mathbb{E}\big[|\mathcal{C}_\alpha(X_{n+1}; \mathcal{D}_{cal})|\big] = \sum_{y \in \mathcal{Y}} \int_{\mathcal{R}} \mathbb{P}\big[B\big(n, F_{cal}(s)\big) \leq n_\alpha\big] p_{S(X_{n+1}, y)}(s) ds \tag{18}$$

$$= \int_{\mathcal{R}} \mathbb{P}\big[B\big(n, F_{cal}(s)\big) \leq n_\alpha\big] \sum_{y \in \mathcal{Y}} p_{S(X_{n+1}, y)}(s) ds, \tag{19}$$

where $\mathcal{R}$ is the space of NC scores and $p_{S(X_{n+1},y)}(s) \triangleq \mathbb{P}\big[S(X_{n+1}, y) = s\big]$. If the NC score is 0-1 loss, i.e. $S(X, Y) = \mathbb{I}\{\hat{Y} \neq Y\}$, we further have $\sum_{y \in \mathcal{Y}} p_{S(X_{n+1},y)}(0) = 1$ and $\sum_{y \in \mathcal{Y}} p_{S(X_{n+1},y)}(1) = |\mathcal{Y}| - 1$. Then

$$\mathbb{E}\big[\big|\mathcal{C}_\alpha(X_{n+1}; \mathcal{D}_{cal})\big|\big] = \mathbb{P}\big[B\big(n, F_{cal}(0)\big) \leq n_\alpha\big] \cdot 1 + \mathbb{P}\big[B\big(n, F_{cal}(1)\big) \leq n_\alpha\big] \cdot \big(|\mathcal{Y}| - 1\big) \quad (20)$$

$$= 1 + \mathbb{P}\big[B\big(n, p_{cal}\big) \leq n_\alpha\big]\big(|\mathcal{Y}| - 1\big). \quad (21)$$

$\square$

## B.3 PROOF OF PROPOSITION 4.2

*Proof.* For symmetric noise define in subsection A.2, we have

$$p_{cal} = \sum_{k=1}^{K} \mathbb{P}_{cal}\big[\hat{Y} = Y, Y = k\big] \quad (22)$$

$$= \sum_{k=1}^{K} \mathbb{P}_{cal}\big[\hat{Y} = k \big| Y = k\big] \mathbb{P}_{cal}[Y = k] \quad (23)$$

$$\approx \sum_{k=1}^{K} \frac{\pi_k m_k(X)}{1 + \sum_{i \neq k} b^{\big(m_k(x) - m_i(x)\big)\big(\frac{K}{K-1}\epsilon - 1\big)}}, \quad (24)$$

where $\pi_k \triangleq \mathbb{P}_{cal}[Y = k]$, $m_k(x) \triangleq \mathbb{P}_{cal}[Y = k | X = x]$ and the derivation of $\mathbb{P}_{cal}\big[\hat{Y} = k \big| Y = k\big]$ in Eq. (24) relies on the analysis of clean accuracy in Theorem 3.3 of (Oyen et al., 2022). The smoothing parameter $b$ controls the sharpness of the softmax approximation to $\arg\max$.

$\square$

Similarly, for asymmetric noise, we also have

$$p_{cal} \approx \sum_{k=1}^{K} \frac{\pi_k m_k(X)}{1 + \sum_{i \neq k} b^{\big(m_k(x) - m_i(x)\big)\big(\frac{s+1}{s}\epsilon - 1\big)}}. \quad (25)$$

Under the assumption of $P[Y = k | X = x] > P[Y = i | X = x](i \neq k)$, $p_{cal}$ decreases as the noise ratio $\epsilon$ increases for both symmetric and asymmetric noise.

## B.4 THE DISTRIBUTION OF HPS AND APS UNDER TRAINING LABEL NOISE

It can be observed that distributions by CE with training label noise can significantly deviate from that of the noise-free case. In Figure 3a, the higher HPS scores under training label noise indicates that the conformity between calibration and test data is poor. In Figure 3b, we show the empirical distributions of APS scores on the clean calibration set with different ratios of training labels. It can be seen that, with the CE baseline, the distribution behaves from being uniform to non-uniform as the noise ratio increases.

As the noise ratio increases, HPS scores for calibration data $\{S_i\}_{i=1}^{n}$ become larger due to the lower conformity with training data in Proposition B.2. So the quantile threshold $\mathcal{Q}_{1-\alpha}\big(\{S_i\}_{i=1}^{n}\big)$ is also larger. For any unlabeled test data $X_{n+1}$, the poor classifier tends to predict a uniform probability for all classes. Then, based on Eq. (1), the prediction set contains more redundant labels. Previous studies (Romano et al., 2020; Einbinder et al., 2022) show that the classifier ia perfect if and only if APS scores follows a uniform distribution for each $X_i$. And we find that APS scores tend to be non-uniform due to the poor classifier under training label noise.

**Proposition B.2.** *Denote $S^\epsilon$ as the corresponding functions defined by the Bayes optimal classifiers trained on noisy training set $\mathcal{D}_{tr}$ with noise ratio $\epsilon$. Assume the clean posterior $P[Y = k | X = x] < \frac{1}{K}$ and $0 \leq \epsilon_1 < \epsilon_2 \leq 1$, then for HPS score of any calibration data $(x, y)$,*

$$S^{\epsilon_1}(x, y) < S^{\epsilon_2}(x, y). \quad (26)$$

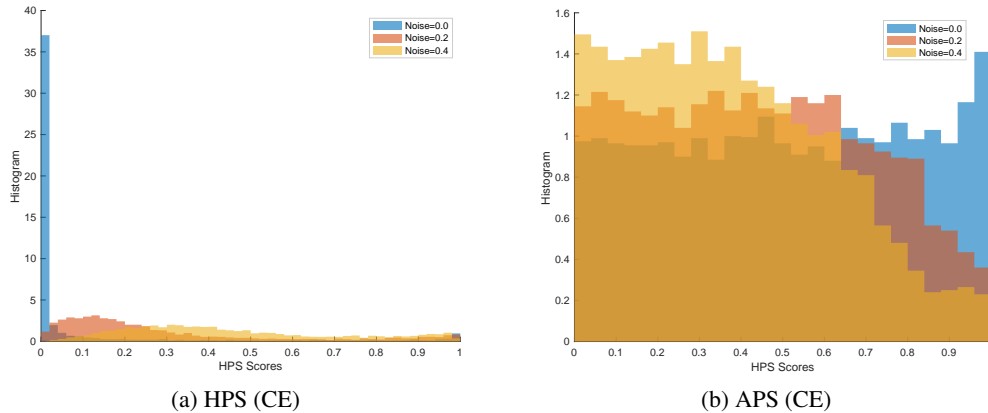

(a) HPS (CE)           (b) APS (CE)

Figure 3: Histogram of HPS and APS scores on CIFAR-10 calibration datasets when training with CE under symmetric training label noise for different noise ratios.

*Proof.* If training labels contain symmetric noise defined in Eq. (15), we have the noisy posterior $\mathbb{P}^\epsilon[Y' = k|X = x]$ satisfies

$$\mathbb{P}^\epsilon[Y' = k|X = x] = \mathbb{P}[Y = k|X = x] - \frac{K\epsilon}{K-1}\mathbb{P}[Y = k|X = x] + \frac{\epsilon}{K-1} \quad (27)$$

$$= \mathbb{P}[Y = k|X = x] + \frac{1 - K\mathbb{P}[Y = k|X = x]}{K-1} \cdot \epsilon, \quad (28)$$

where Eq. (27) is derived by Oyen et al. (2022) in Eq. (5) in their work. Under the assumption of the clean posterior $\mathbb{P}[Y = k|X = x] < \frac{1}{K}$ for any noisy label $k \in \mathcal{Y}\backslash Y$, if $0 \leq \epsilon_1 < \epsilon_2 \leq 1$, then

$$S^{\epsilon_1}(x, k) = 1 - \mathbb{P}^{\epsilon_1}[Y' = k|X = x] > \mathbb{P}^{\epsilon_2}[Y' = k|X = x] = S^{\epsilon_2}(x, k). \quad (29)$$

For true label $Y = y$,

$$S^{\epsilon_1}(x, y) = 1 - \sum_{k \in \mathcal{Y}\backslash y} S^{\epsilon_1}(x, k) < 1 - \sum_{k \in \mathcal{Y}\backslash y} S^{\epsilon_2}(x, k) = S^{\epsilon_2}(x, y). \quad (30)$$

$\square$

## C   More analysis of Generalization and convergence

### C.1   Proof of Theorem 5.1

*Proof.* Define:

$$\epsilon \quad := \quad \frac{3}{\sqrt{M}}, \quad (31)$$

$$\Delta \quad := \quad \frac{\sqrt{2d\ln(3/\epsilon) + 2\ln(2/\delta)}}{\sqrt{M}}. \quad (32)$$

Using Hoeffding's inequality we have for any fixed $\Theta$,

$$P\big\{\big|\hat{R}(\hat{\mathbf{w}}(\Theta)) - R(\hat{\mathbf{w}}(\Theta))\big| > \Delta\big\} \leq 2\exp\left(-\frac{M\Delta^2}{2}\right) = \frac{\delta}{(3/\epsilon)^d}. \quad (33)$$

Let $\mathcal{A}$ be an $\epsilon$-cover of $\mathbb{B}^d$, we have $|\mathcal{A}| \leq (1 + 2/\epsilon)^d$. By the assumption of $\epsilon \leq 1$, then $|\mathcal{A}| \leq (3/\epsilon)^d$. Using the above and union bounding over all elements of $\mathcal{A}$, we have

$$P\big\{\forall\Theta \in \mathcal{A} : \big|\hat{R}(\hat{\mathbf{w}}(\Theta)) - R(\hat{\mathbf{w}}(\Theta))\big| \leq \Delta\big\} \geq 1 - \delta, \quad (34)$$

*i.e.*, for any $\Theta$ in $\mathcal{A}$, we have

$$\left|\hat{R}(\hat{\mathbf{w}}(\Theta)) - R(\hat{\mathbf{w}}(\Theta))\right| \leq \sqrt{\frac{2d\ln(3/\epsilon) + 2\ln(2/\delta)}{M}}. \tag{35}$$

Thus, for any $\Theta'$ in $\mathcal{A}$,

$$R(\hat{\mathbf{w}}(\hat{\Theta})) \leq \hat{R}(\hat{\mathbf{w}}(\hat{\Theta})) + \sqrt{\frac{2d\ln(3/\epsilon) + 2\ln(2/\delta)}{M}} \tag{36}$$

$$\leq \hat{R}(\hat{\mathbf{w}}(\Theta')) + \sqrt{\frac{2d\ln(3/\epsilon) + 2\ln(2/\delta)}{M}} \tag{37}$$

$$\leq R(\hat{\mathbf{w}}(\Theta')) + 2\sqrt{\frac{2d\ln(3/\epsilon) + 2\ln(2/\delta)}{M}}. \tag{38}$$

As $\hat{\Theta}, \Theta' \in \mathcal{A}$, we get Eq. (36) and Eq. (38) use the bound in Eq. (35), and Eq. (37) is by the definition of $\hat{\Theta}$, i.e. $\hat{\Theta} = \underset{\Theta \in \mathcal{A}}{\arg\min} \hat{R}(\hat{\mathbf{w}}(\Theta))$. Then, under the assumption $\Omega(X^{meta}; \hat{\mathbf{w}}(\Theta))$ is $\lambda$-*Lipschitz* continuous w.r.t. $\Theta$, $\forall \Theta \in \mathbb{B}^d$, we have

$$R(\hat{\mathbf{w}}(\Theta)) \geq R(\hat{\mathbf{w}}(\Theta')) - \lambda\epsilon \tag{39}$$

$$\geq R(\hat{\mathbf{w}}(\hat{\Theta})) - 2\sqrt{\frac{2d\ln(3/\epsilon) + 2\ln(2/\delta)}{M}} - \lambda\epsilon \tag{40}$$

$$\geq R(\hat{\mathbf{w}}(\hat{\Theta})) - \frac{3\lambda + \sqrt{4d\ln(M) + 8\ln(2/\delta)}}{\sqrt{M}}. \tag{41}$$

For $\Theta^* = \underset{\Theta \in \mathbb{B}^d}{\arg\min} R(\hat{\mathbf{w}}(\Theta))$ be the optimal parameter in the unit ball, obviously,

$$R(\hat{\mathbf{w}}(\hat{\Theta})) - R(\hat{\mathbf{w}}(\Theta^*)) \leq \frac{3\lambda + \sqrt{4d\ln(M) + 8\ln(2/\delta)}}{\sqrt{M}}. \tag{42}$$

$$\square$$

## C.2 THE CONVERGENCE GUARANTEE OF CONF-MWN

We can provide the convergence guarantee for the training loss and the meta loss based on Theorem 1 and Theorem 2 in CE-MWN (Shu et al., 2019). All we need to do is to verify the Lipschitz smoothness of the meta loss.

**Assumption C.1** (Lipschitz Smoothness). Assume that the k-th output value $f_k$ of the classifier is Lipschitz smooth with respect to the classifier parameters $\hat{\mathbf{w}}^{(t)}(\Theta)$, then $\Omega_{\alpha,j_2}(\hat{\mathbf{w}}^{(t)}(\Theta))$ is also Lipschitz smooth with respect to the classifier parameters $\hat{\mathbf{w}}^{(t)}(\Theta)$.

For convenience, let $\nabla f_k(\hat{\mathbf{w}}^{(t)}(\Theta)) = \frac{\partial f_k}{\partial \hat{\mathbf{w}}^{(t)}(\Theta)}$. By the Lipschitz smoothness of $f_k$, there exists a constant $L_1$ such that for any $\hat{\mathbf{w}}^{(t)}(\Theta)_1$ and $\hat{\mathbf{w}}^{(t)}(\Theta)_2$, we have

$$\|\nabla f_k(\hat{\mathbf{w}}^{(t)}(\Theta)_1) - \nabla f_k(\hat{\mathbf{w}}^{(t)}(\Theta)_2)\|_2 \leq L_1\|\hat{\mathbf{w}}^{(t)}(\Theta)_1 - \hat{\mathbf{w}}^{(t)}(\Theta)_2\|_2. \tag{43}$$

Since $\sigma(x) \in (0, 1), \forall x \in \mathbb{R}$, then for any $\hat{\mathbf{w}}^{(t)}(\Theta)$, $x_j$ and $y$,

$$\sigma\left(\frac{\widetilde{Q}_{1-\alpha}(\hat{\mathbf{w}}^{(t)}(\Theta) - S(x_j, y))}{T}\right)\left[1 - \sigma\left(\frac{\widetilde{Q}_{1-\alpha}(\hat{\mathbf{w}}^{(t)}(\Theta) - S(x_j, y))}{T}\right)\right] \leq \frac{1}{4}. \tag{44}$$

So there exists $L_2 = \frac{K}{4T}L_1$ such that

$$\|\nabla\Omega_{\alpha,j_2}(\hat{\mathbf{w}}^{(t)}(\Theta)_1) - \nabla\Omega_{\alpha,j_2}(\hat{\mathbf{w}}^{(t)}(\Theta)_2)\|_2 \leq \frac{K}{4T} \times \|\nabla f_k(\hat{\mathbf{w}}^{(t)}(\Theta)_1) - \nabla f_k(\hat{\mathbf{w}}^{(t)}(\Theta)_2)\|_2$$

$$\leq L_2\|\hat{\mathbf{w}}^{(t)}(\Theta)_1 - \hat{\mathbf{w}}^{(t)}(\Theta)_2\|_2. \tag{45}$$

## C.3 THE EMPIRICAL CONVERGENCE OF CONF-MWN

We plot the empirical loss convergence curves for Conf-MWN both in the upper level and lower level with different random seeds in Figure 4. It can be seen that both the inner and outer optimizations emprically tend to converge.

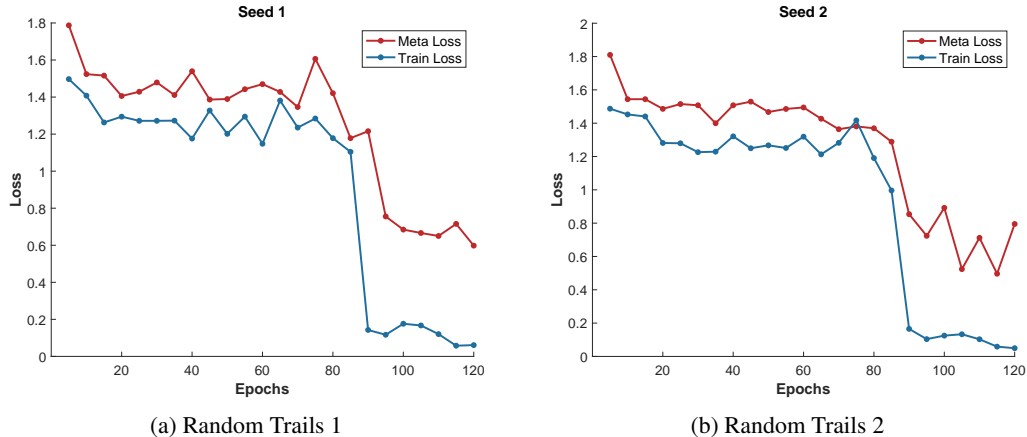

(a) Random Trails 1          (b) Random Trails 2

Figure 4: The convergence of Train Loss and Meta Loss for our proposed Conf-MWN method on CIFAR-10 datasets with 20% symmetric noise.

# D  MORE EXPERIMENTAL DETAILS

## D.1 CIFAR-10 AND CIFAR-100 DATASETS WITH SYNTHETIC LABEL NOISE

**Models and training settings.** For both CIFAR-10 and CIFAR-100 datasets, we adopt ResNet-32 (He et al., 2016) as the classifier, and an MLP with one hidden layer as the weighting function. Both the classifier and weight networks are trained using SGD with a momentum of 0.9, a weight decay of $5 \times 10^{-4}$, and an initial learning rate of 0.1. The learning rate of the classifier is divided by 10 after 80 epochs and 100 epochs (for total 120 epochs). The learning rate of weight-net is fixed as $10^{-3}$ in the meta-training stage. We repeated all experiments 3 times with different random seeds. To better compare the results, we follows the experimental design of CE-MWN (Shu et al., 2019).

## D.2 CIFAR-10N, CIFAR-100N AND FOOD-101N DATASETS WITH REAL-WORLD NOISE

**Datasets.** We randomly split 1000 data from the training datasets with their true labels got from the original CIFAR-10 training set as meta data, and randomly split 5000 the test datasets as calibration data. For CIFAR-100N datasets, the meta data and the calibration data are obtained in a similar way as that for CIFAR-10N. Food-101N datasets contain about 310K images of food recipes classified into 101 class and about 20% of the data are mislabeled. 25K images with curated annotations are selected from Food-101 datasets (Bossard et al., 2014) as the test set. We split 2020 images from it as the meta data.

**Models and training settings.** For CIFAR-10N and CIFAR-100N datasets, we use the same models and training settings as in Section 6.1. For Food-101N, we adopt ResNet-50 (pre-trained on ImageNet) as the classifier network. On the first 5 epochs, we directly train the classifier network with CE for all comparison methods, while on later epochs (a total of 30 epochs), Conf-MWN and CE-MWN are trained with the meta-learning model. All the networks were trained using SGD with a weight decay of $1 \times 10^{-3}$ and an initial learning rate of $5 \times 10^{-3}$. The learning rate in the meta-training stage is fixed as $10^{-3}$ for both Conf-MWN and CE-MWN. We repeat the experiments 3 times with different random seeds for CIFAR-10N and CIFAR-100N datasets, as well as 5 times for Food-101N datasets.

# E   ADDITIONAL EXPERIMENTAL RESULTS

## E.1   OTHER ERROR RATES

In addition to setting the error rate to $\alpha = 0.01$ for 99.00% coverage in our main experiments, we also test with $\alpha$ taking 0.05 and 0.1 on CIFAR-10 and CIFAR-100 datasets with synthetic noise. The results in Tables 5 and 6 show the efficiency of our method in reducing the set size under training label noise.

Table 5: Results on CIFAR-10 datasets with symmetric (sym.) noise for $\alpha = 0.5$ or $\alpha = 0.1$. The performance is averaged over 3 random runs and the best is highlighted in **Bold**.

|  | $\alpha$=0.05 | | | | $\alpha$=0.1 | | | |
| --- | --- | --- | --- | --- | --- | --- | --- | --- |
|  | Sym.-20% | | Sym.-40% | | Sym.-20% | | Sym.-40% | |
| Methods | Cov.% | Ineff. | Cov.% | Ineff. | Cov.% | Ineff. | Cov.% | Ineff. |
| CE | 95.39 | 1.46 | 95.03 | 2.16 | 90.64 | 1.09 | 90.43 | 1.28 |
| Conf-MWN | 95.14 | **1.19** | 95.12 | **1.14** | 90.57 | **1.01** | 90.29 | **1.10** |

Table 6: Results on CIFAR-100 datasets for $\alpha = 0.1$ with different symmetric (sym.) noise ratios. The performance is averaged over 3 random runs and the best is highlighted in **Bold**.

|  | Sym.-20% | | Sym.-40% | | Sym.-60% | | Sym.-80% | |
| --- | --- | --- | --- | --- | --- | --- | --- | --- |
| Methods | Cov.% | Ineff. | Cov.% | Ineff. | Cov.% | Ineff. | Cov.% | Ineff. |
| CE | 90.86 | 5.64 | 90.85 | 9.40 | 90.63 | 18.93 | 90.71 | 48.32 |
| Conf-MWN | 91.14 | **4.28** | 91.44 | **6.33** | 90.53 | **11.80** | 90.30 | **47.82** |

## E.2   THE PREDICTION SET SIZES WITH APS, RAPS AND SAPS SCORES

We further show the results of APS (Romano et al., 2020), RAPS (Angelopoulos et al., 2021) and SAPS (Huang et al., 2023) (with the optimal parameters) under the condition of alpha = 0.01 in Tables 7 and 8. We find that the set size is affected by training label noise, regardless of which NC score is used, and Conf-WMN could improve the efficiency of the prediction set significantly.

Table 7: Results on CIFAR-10 datasets with symmetric (sym.) noise for different NC scores at $\alpha = 0.01$. The performance is averaged over 3 random runs and the smallest size is highlighted in **Bold**.

|  | Sym.-20% | | | | Sym.-40% | | | |
| --- | --- | --- | --- | --- | --- | --- | --- | --- |
|  | Cov.% | | Ineff. | | Cov.% | | Ineff. | |
| Scores | CE | Conf-MWN | CE | Conf-MWN | CE | Conf-MWN | CE | Conf-MWN |
| APS | 99.98 | 98.69 | 6.30 | **2.35** | 99.25 | 98.74 | 7.57 | **3.39** |
| RAPS | 99.00 | 98.76 | 6.17 | **2.22** | 99.15 | 98.73 | 7.44 | **3.00** |
| SAPS | 98.80 | 98.67 | 7.42 | **3.01** | 98.85 | 98.67 | 8.39 | **3.71** |

## E.3   THE RESULTS OF ASYMMETRIC NOISE

Tables 9 and 10 summarize the results with asymmetric noise on CIFAR-10 and CIFAR-100 datasets, respectively. The effectiveness of the proposed method in improving the efficiency of the prediction sets can be observed.

Table 8: Results on CIFAR-100 datasets with symmetric (sym.) noise for different NC scores at $\alpha = 0.01$. The performance is averaged over 3 random runs and the smallest size is highlighted in **Bold**.

| | Sym.-20% | | | | Sym.-40% | | | |
| | Cov.% | | Ineff. | | Cov.% | | Ineff. | |
| Scores | CE | Conf-MWN | CE | Conf-MWN | CE | Conf-MWN | CE | Conf-MWN |
| --- | --- | --- | --- | --- | --- | --- | --- | --- |
| APS | 98.81 | 98.95 | 43.27 | **29.66** | 98.74 | 98.88 | 57.88 | **40.69** |
| RAPS | 98.88 | 98.93 | 42.61 | **27.46** | 98.80 | 98.84 | 58.60 | **37.38** |
| SAPS | 98.88 | 99.08 | 42.20 | **31.79** | 98.89 | 99.17 | 55.76 | **36.73** |

Table 9: Results on CIFAR-10 datasets with asymmetric (asy.) noise. The performance is averaged over 3 random runs and the best is highlighted in **Bold**.

| | Asy.-20% | | | Asy.-40% | | |
| Methods | Cov.% | Ineff. | Acc.% | Cov.% | Ineff. | Acc.% |
| --- | --- | --- | --- | --- | --- | --- |
| CE | 98.96 | 2.84 | 89.43 | 98.87 | 2.91 | 81.34 |
| CE-MWN | 98.93 | 2.88 | 90.26 | 98.85 | 3.45 | 86.16 |
| Conf-MWN | 98.98 | **2.74** | 90.48 | 98.76 | **2.80** | 83.19 |

Table 10: Results on CIFAR-100 datasets with asymmetric (asy.) noise. The performance is averaged over 3 random runs and the best is highlighted in **Bold**.

| | Asy.-20% | | | Asy.-40% | | |
| Methods | Cov.% | Ineff. | Acc.% | Cov.% | Ineff. | Acc.% |
| --- | --- | --- | --- | --- | --- | --- |
| CE | 98.82 | 22.72 | 63.53 | 98.86 | 27.02 | 48.98 |
| CE-MWN | 98.74 | 21.72 | 64.45 | 98.88 | 25.76 | 50.39 |
| Conf-MWN | 98.82 | **21.14** | 64.32 | 98.88 | **25.10** | 50.73 |

### E.4 THE DISTRIBUTION OF HPS AND APS USING CONF-MWN

See Figure 5. Compared with Figure 3a, when using our method for training, most of the HPS scores are concentrated around small values, even when the noise ratio is 40%. And APS scores tend to be uniform as we expected.

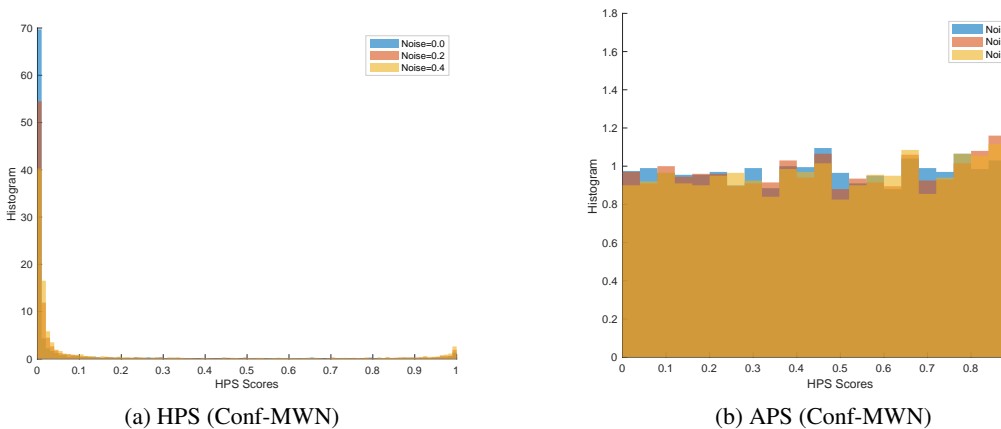

(a) HPS (Conf-MWN)                    (b) APS (Conf-MWN)

Figure 5: Histogram of HPS and APS scores on CIFAR-10 calibration datasets when training using Conf-MWN under symmetric training label noise for different noise ratios.

### E.5 COMPARISONS WITH OTHER ROBUST LEARNING METHODS

See Tables 11 and 12.

Table 11: Results on CIFAR-10 datasets with symmetric (sym.) noise at $\alpha = 0.01$. The performance is averaged over 3 random runs and the best is highlighted in **Bold**.

| Methods | Sym.-20% | | | Sym.-40% | | |
|---|---|---|---|---|---|---|
| | Cov.% | Ineff. | Acc.% | Cov.% | Ineff. | Acc.% |
| CE | 98.95 | 5.31 | 87.55 | 99.03 | 7.42 | 83.78 |
| CE-MWN | 98.98 | 2.66 | 89.78 | 98.81 | 3.03 | 86.62 |
| GCE | 98.45 | 2.08 | 89.06 | 98.92 | 3.18 | 86.99 |
| Co-teaching | 98.74 | 3.35 | 78.40 | 98.60 | 4.61 | 74.45 |
| Conf-MWN | 98.82 | **1.98** | 90.18 | 98.99 | **2.85** | 87.08 |

Table 12: Results on CIFAR-100 datasets with symmetric (sym.) noise at $\alpha = 0.01$. The performance is averaged over 3 random runs and the best is highlighted in **Bold**.

| Methods | Sym.-20% | | | Sym.-40% | | |
|---|---|---|---|---|---|---|
| | Cov.% | Ineff. | Acc.% | Cov.% | Ineff. | Acc.% |
| CE | 99.01 | 40.69 | 62.23 | 98.96 | 58.67 | 55.71 |
| CE-MWN | 98.94 | 27.15 | 64.68 | 98.86 | 42.09 | 58.32 |
| GCE | 98.90 | 31.86 | 61.06 | 99.03 | 42.78 | 56.13 |
| Co-teaching | 98.88 | 36.48 | 44.65 | 99.01 | 41.11 | 42.67 |
| Conf-MWN | 99.03 | **24.50** | 65.44 | 98.91 | **33.60** | 60.20 |

### E.6 COMPARISONS WITH CONFORMAL TRAINING METHODS

See Table 13.

Table 13: Results on CIFAR-10 datasets with symmetric (sym.) noise at $\alpha = 0.01$. The performance is averaged over 3 random runs and the best is highlighted in **Bold**.

| Methods | Sym-20% | | | Sym-40% | | |
|---|---|---|---|---|---|---|
| | Cov.% | Ineff. | Acc.% | Cov.% | Ineff. | Acc.% |
| CE | 98.95 | 5.31 | 87.55 | 99.03 | 7.42 | 83.78 |
| ConfTr Size Loss | 99.12 | 8.08 | 33.28 | 99.44 | 7.96 | 39.03 |
| CE+0.1×ConfTr Size Loss | 98.86 | 5.08 | 88.13 | 99.19 | 7.30 | 84.22 |
| CE+1e$^{-5}$×ConfTr Size Loss | 99.24 | 6.09 | 87.87 | 98.83 | 7.23 | 82.94 |
| Conf-MWN | 98.82 | **1.98** | 90.18 | 98.99 | **2.85** | 87.08 |

## E.7 BOTH TRAINING AND CALIBRATION DATA CONTAIN LABEL NOISE

See Tables 14, 15 and 16.

Table 14: Results on CIFAR-10 datasets with both training and calibration sets containing label noise ($\alpha = 0.01$). The best performance is highlighted in **Bold**.

| Scores | Sym.-20% | | | | Sym.-40% | | | |
|---|---|---|---|---|---|---|---|---|
| | Cov.% | | Ineff. | | Cov.% | | Ineff. | |
| | CE | Conf-MWN | CE | Conf-MWN | CE | Conf-MWN | CE | Conf-MWN |
| HPS | 99.95 | 99.97 | 9.48 | 9.19 | 99.92 | 99.95 | 9.68 | 9.58 |
| NACP+HPS | 99.21 | 99.35 | 6.85 | **2.56** | 99.33 | 99.26 | 8.20 | **5.40** |
| APS | 99.96 | 99.98 | 9.57 | 9.38 | 99.93 | 100.0 | 9.80 | 9.72 |
| NACP+APS | 99.58 | 99.72 | 7.90 | **4.71** | 99.15 | 99.53 | 7.41 | **6.71** |

Table 15: Results on CIFAR-10 datasets with both training and calibration sets containing label noise ($\alpha = 0.1$). The best performance is highlighted in **Bold**.

| Scores | Sym.-20% | | | | Sym.-40% | | | |
|---|---|---|---|---|---|---|---|---|
| | Cov.% | | Ineff. | | Cov.% | | Ineff. | |
| | CE | Conf-MWN | CE | Conf-MWN | CE | Conf-MWN | CE | Conf-MWN |
| HPS | 98.87 | 99.73 | 5.53 | 4.50 | 99.24 | 99.93 | 7.62 | 6.94 |
| NACP-HPS | 91.62 | 91.92 | 1.15 | **1.05** | 92.14 | 92.58 | 1.51 | **1.20** |
| APS | 98.70 | 99.70 | 5.52 | 4.74 | 99.41 | 99.90 | 7.79 | 7.08 |
| NACP-APS | 90.41 | 89.72 | 1.86 | **1.18** | 91.56 | 91.93 | 2.57 | **1.44** |

## E.8 DISCUSSIONS OF CONDITIONAL COVERAGE

Define $\{\mathcal{C}_j\}_{j=1}^s$ as disjoint size strata partitioning $\{1, \ldots, |\mathcal{Y}|\}$ and index sets $\mathcal{J}_j = \{i : |\mathcal{C}_\alpha(X_i; \mathcal{D}_{cal})| \in \mathcal{C}_j\}$ grouping examples by prediction set size. Then SSCV (Angelopoulos et al., 2021) is defined over these strata as on strata $\{\mathcal{C}_j\}_{j=1}^s$ as

$$\text{SSCV}(\mathcal{C}_\alpha, \{\mathcal{C}_j\}_{j=1}^s) = \sup_j \left| \frac{|i : Y_i \in \mathcal{C}_\alpha(X_i; \mathcal{D}_{cal}), i \in \mathcal{J}_j|}{|\mathcal{J}_j|} - (1 - \alpha) \right|. \quad (46)$$

And we denote $\mathcal{J}^y = \{i : Y_i = y\}$ as the indices of test examples with label $y$ and $\hat{c}_y = \frac{1}{|\mathcal{J}^y|} \sum_{i \in \mathcal{J}^y} \mathbb{I}\{Y_i \in \mathcal{C}_\alpha(X_i; \mathcal{D}_{cal})\}$ as the empirical class-conditional coverage of class $y$. Then CovGap (Ding et al., 2023) is defined as:

$$\text{CovGap} = 100 \times \frac{1}{|\mathcal{Y}|} \sum_{y \in \mathcal{Y}} |\hat{c}_y - (1 - \alpha)|. \quad (47)$$

Table 16: Results on CIFAR-100 datasets with both training and calibration sets containing label noise ($\alpha = 0.1$). The best performance is highlighted in **Bold**.

| | Sym.-20% | | | | Sym.-40% | | | |
| | Cov.% | | Ineff. | | Cov.% | | Ineff. | |
| Scores | CE | Conf-MWN | CE | Conf-MWN | CE | Conf-MWN | CE | Conf-MWN |
|---|---|---|---|---|---|---|---|---|
| HPS | 100.0 | 100.0 | 93.05 | 94.49 | 99.99 | 100.0 | 96.16 | 96.72 |
| NACP-HPS | 99.64 | 99.44 | 62.51 | **34.73** | 98.86 | 99.74 | 56.97 | **40.24** |
| APS | 100.0 | 100.0 | 94.14 | 94.58 | 99.99 | 100.0 | 96.39 | 96.30 |
| NACP-APS | 99.47 | 99.39 | 58.45 | 38.34 | 99.14 | 98.88 | 66.52 | 40.94 |

To measure SSCV on CIFAR-10 and CIFAR-100 datasets, we divide prediction sets into $s = 5$ groups based on sizes: 0-1, 2-3, 4-10, 11-100 and 101-1000. The results in terms of these two metrics are summarized in Tables 17, 18, 19 and Tables 20, 21, 22. As can be seen, our method has comparable results to the CE baseline, showing that it can maintain a reasonable conditional coverage in general, though not its main focus.

Table 17: Comparisons between CE and Conf-MWN in terms of CovGap (%) ↓ and SSCV ↓ for HPS and APS at $\alpha = 0.01$ on CIFAR-10 datasets.

| | No Noise | | | | Sym-20% | | | | Sym-40% | | | |
| | CovGap | | SSCV | | CovGap | | SSCV | | CovGap | | SSCV | |
| Methods | HPS | APS | HPS | APS | HPS | APS | HPS | APS | HPS | APS | HPS | APS |
|---|---|---|---|---|---|---|---|---|---|---|---|---|
| CE | 0.54 | 0.53 | 0.01 | 0.01 | 0.47 | 0.50 | 0.02 | 0.03 | 0.37 | 0.50 | 0.03 | 0.04 |
| Conf-MWN | 0.55 | 0.50 | 0.01 | 0.01 | 0.53 | 0.66 | 0.00 | 0.00 | 0.49 | 0.63 | 0.01 | 0.01 |

Table 18: Comparisons between CE and Conf-MWN in terms of CovGap (%) ↓ and SSCV ↓ for HPS and APS at $\alpha = 0.05$ on CIFAR-10 datasets.

| | No Noise | | | | Sym-20% | | | | Sym-40% | | | |
| | CovGap | | SSCV | | CovGap | | SSCV | | CovGap | | SSCV | |
| Methods | HPS | APS | HPS | APS | HPS | APS | HPS | APS | HPS | APS | HPS | APS |
|---|---|---|---|---|---|---|---|---|---|---|---|---|
| CE | 2.17 | 0.99 | 0.07 | 0.01 | 1.92 | 1.64 | 0.04 | 0.04 | 1.86 | 1.86 | 0.01 | 0.05 |
| Conf-MWN | 2.43 | 1.23 | 0.05 | 0.01 | 2.15 | 1.35 | 0.05 | 0.02 | 2.37 | 1.43 | 0.02 | 0.02 |

### E.9 ADDITIONAL RESULTS ON DATASETS WITH REAL-WORLD LABEL NOISE

Tables 23, 24 and 25 show more experimental results on CIFAR-10N, CIFAR-100N and Food-101N datasets. These results further substantiate the effectiveness of the proposed method.

Table 19: Comparisons between CE and Conf-MWN in terms of CovGap (%) ↓ and SSCV ↓ for HPS and APS at $\alpha = 0.1$ on CIFAR-10 datasets.

| Methods | No Noise | | | | Sym-20% | | | | Sym-40% | | | |
| | CovGap | | SSCV | | CovGap | | SSCV | | CovGap | | SSCV | |
| | HPS | APS | HPS | APS | HPS | APS | HPS | APS | HPS | APS | HPS | APS |
|---|---|---|---|---|---|---|---|---|---|---|---|---|
| CE | 3.86 | 1.56 | 0.01 | 0.03 | 3.64 | 2.32 | 0.07 | 0.05 | 3.58 | 2.49 | 0.06 | 0.05 |
| Conf-MWN | 4.54 | 1.03 | 0.01 | 0.08 | 4.01 | 1.76 | 0.01 | 0.01 | 4.36 | 2.39 | 0.10 | 0.03 |

Table 20: Comparisons between CE and Conf-MWN in terms of CovGap (%) ↓ and SSCV ↓ for HPS and APS at $\alpha = 0.01$ on CIFAR-100 datasets.

| Methods | No Noise | | | | Sym-20% | | | | Sym-40% | | | |
| | CovGap | | SSCV | | CovGap | | SSCV | | CovGap | | SSCV | |
| | HPS | APS | HPS | APS | HPS | APS | HPS | APS | HPS | APS | HPS | APS |
|---|---|---|---|---|---|---|---|---|---|---|---|---|
| CE | 0.62 | 0.85 | 0.01 | 0.00 | 0.77 | 0.95 | 0.00 | 0.03 | 0.80 | 0.82 | 0.01 | 0.03 |
| Conf-MWN | 0.81 | 0.77 | 0.01 | 0.00 | 0.83 | 0.84 | 0.01 | 0.01 | 0.82 | 0.86 | 0.01 | 0.01 |

Table 21: Comparisons between CE and Conf-MWN in terms of CovGap (%) ↓ and SSCV ↓ for HPS and APS at $\alpha = 0.05$ on CIFAR-100 datasets.

| Methods | No Noise | | | | Sym-20% | | | | Sym-40% | | | |
| | CovGap | | SSCV | | CovGap | | SSCV | | CovGap | | SSCV | |
| | HPS | APS | HPS | APS | HPS | APS | HPS | APS | HPS | APS | HPS | APS |
|---|---|---|---|---|---|---|---|---|---|---|---|---|
| CE | 2.34 | 2.27 | 0.05 | 0.03 | 2.57 | 2.13 | 0.04 | 0.04 | 2.65 | 2.22 | 0.03 | 0.07 |
| Conf-MWN | 2.44 | 2.23 | 0.06 | 0.02 | 2.57 | 2.16 | 0.04 | 0.01 | 2.88 | 2.40 | 0.04 | 0.03 |

Table 22: Comparisons between CE and Conf-MWN in terms of CovGap (%) ↓ and SSCV ↓ for HPS and APS at $\alpha = 0.1$ on CIFAR-100 datasets.

| Methods | No Noise | | | | Sym-20% | | | | Sym-40% | | | |
| | CovGap | | SSCV | | CovGap | | SSCV | | CovGap | | SSCV | |
| | HPS | APS | HPS | APS | HPS | APS | HPS | APS | HPS | APS | HPS | APS |
|---|---|---|---|---|---|---|---|---|---|---|---|---|
| CE | 4.30 | 2.74 | 0.16 | 0.02 | 4.48 | 3.00 | 0.11 | 0.06 | 4.53 | 3.42 | 0.08 | 0.11 |
| Conf-MWN | 3.86 | 2.95 | 0.16 | 0.02 | 4.28 | 3.34 | 0.11 | 0.03 | 4.49 | 3.80 | 0.08 | 0.04 |

Table 23: Results on CIFAR-10N datasets with real noise. The performance is averaged over 3 random runs and the best is highlighted in **Bold**. **Green** represents the reduction in Ineff. compared to CE.

| Methods | Random 2-18.12% | | | Random 3-17.64% | | |
| | Cov.% | Ineff. | Acc.% | Cov.% | Ineff. | Acc.% |
|---|---|---|---|---|---|---|
| CE | 99.02 | 4.44 | 87.57 | 98.94 | 3.84 | 87.51 |
| CE-MWN | 98.94 | 2.51 (-43.47%) | 89.27 | 98.93 | 2.52 (-34.38%) | 89.08 |
| Conf-MWN | 98.99 | **2.47** (-44.37%) | 89.41 | 99.07 | **2.45** (-36.20%) | 88.83 |

Table 24: Results on CIFAR-100N datasets with real noise. The performance is averaged over 3 random runs and the best is highlighted in **Bold**. **Green** represents the reduction compared to CE.

| Methods | Fine Label-40.20% | | |
|---|---|---|---|
| | Cov.% | Ineff. | Acc.% |
| CE | 98.95 | 34.28 | 54.43 |
| CE-MWN | 99.13 | 31.40 (-8.40%) | 56.35 |
| Conf-MWN | 98.99 | **28.03** (-18.23%) | 56.56 |

Table 25: Results on Food-101N datasets with real noise. The performance is averaged over 5 random runs and the best is highlighted in **Bold**. **Green** represents the reduction compared to CE.

| Methods | ≈ 20% | | |
|---|---|---|---|
| | Cov.% | Ineff. | Acc.% |
| CE | 99.01 | 14.28 | 78.54 |
| CE-MWN | 99.00 | 8.17 (-42.79%) | 85.27 |
| Conf-MWN | 99.00 | **7.96** (-44.26%) | 85.30 |

# F EFFECTS OF HYPERPARAMETERS

## F.1 DIFFERENT CLASSIFIERS

We conduct experiments with different networks, including ResNet-18, ResNet-34, and ResNet-50 (pre-trained on ImageNet), on CIFAR-10 with symmetric noise. The results in terms of `Ineff.` and `Acc.` with noise ratio 20% and 40% against CE are shown in Figure 6. It can be seen that our method can consistently improve the `Ineff.` over CE with different networks. Besides, as the model architecture becomes more complex, CE tends to produce worse results, indicating that it overfits the noisy training data, and our method can well alleviate this issue.

## F.2 THE NUMBER OF CLEAN META DATA

In Figure 7, we show the results of the set sizes on CIFAR-10N datasets by varying the number of clean meta data, which are randomly selected. It can be seen that, even with a very small amount of meta data, i.e., 100, the performance improvement over the CE baseline can be observed.

## F.3 THE TEMPERATURE $T$ IN META LOSS

We also analyze on CIFAR-10 datasets with 20% symmetric label noise by varying the temperature $T$ in the differentiable relaxation of the meta loss, and the results are summarized in Table 26. It can be seen that the results are not very sensitive to this hyperparameter when it is not too large ($T \leq 1$). We also show the convergence curve of training loss and meta loss with different $T$ in Figure 8.

Table 26: Results of Conf-MWN by varying temperature $T$ on CIFAR-10 datasets with 20% symmetric noise.

| T | 0.01 | 0.1 | 0.5 | 1 | 5 | 10 |
|---|---|---|---|---|---|---|
| Ineff. | 2.04 | 1.97 | 2.01 | 1.98 | 2.49 | 2.30 |
| Cov.% | 98.91 | 98.95 | 98.98 | 98.82 | 99.09 | 98.89 |
| Acc.% ↑ | 90.34 | 90.16 | 90.27 | 90.18 | 90.20 | 89.69 |

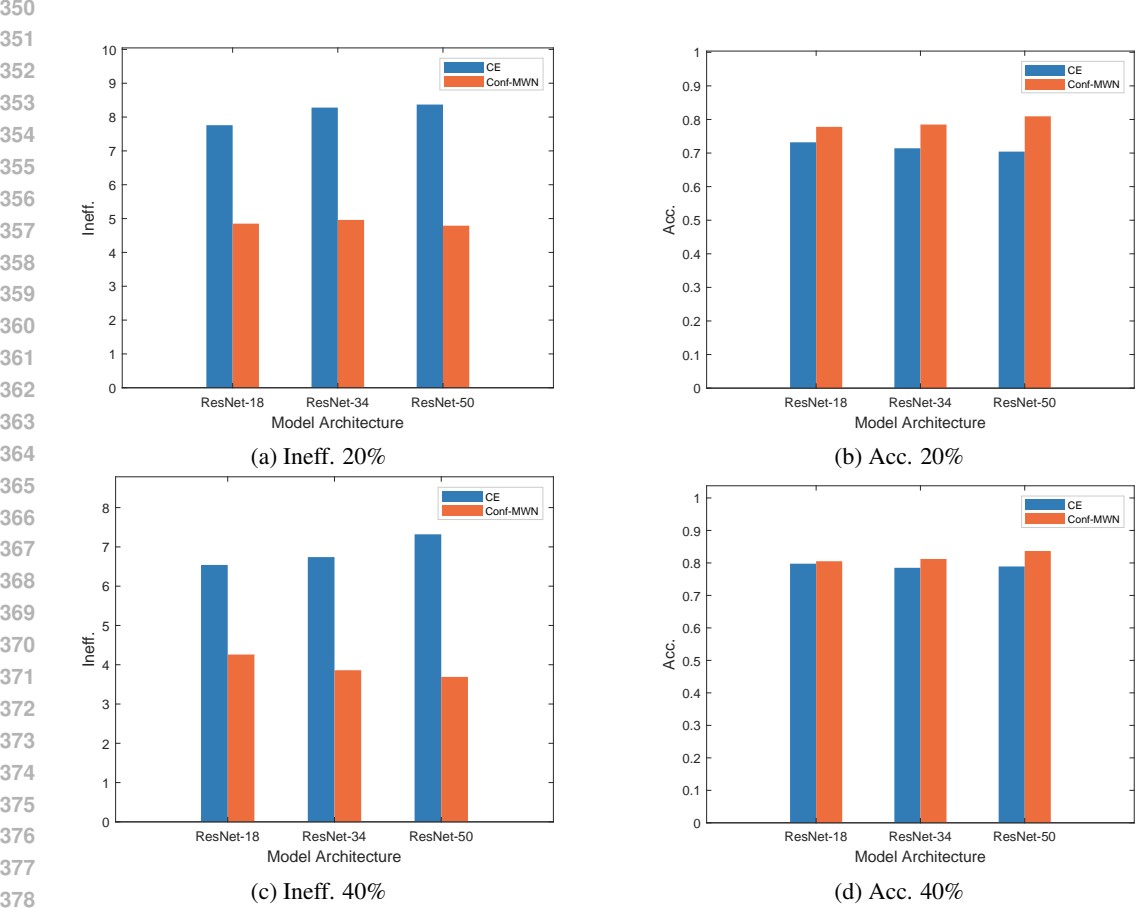

(a) Ineff. 20%

(b) Acc. 20%

(c) Ineff. 40%

(d) Acc. 40%

Figure 6: `Ineff.` and `Acc.` on CIFAR-10 datasets with 20% and 40% symmetric noise using different classifiers.

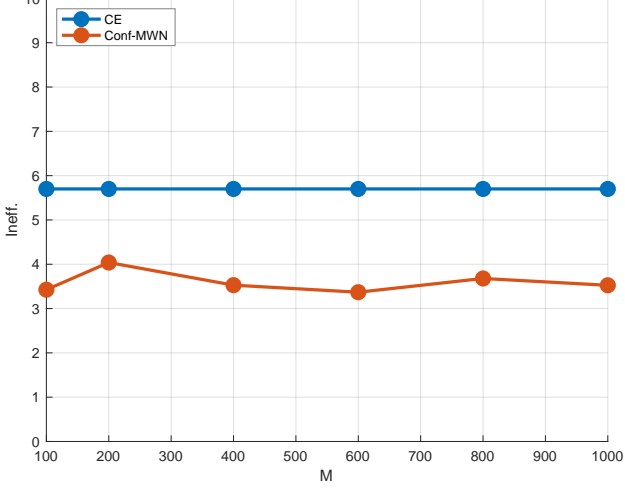

Figure 7: Inefficiency results of Conf-MWN by varying the number of meta data on CIFAR-10N datasets ("Worst" noise).

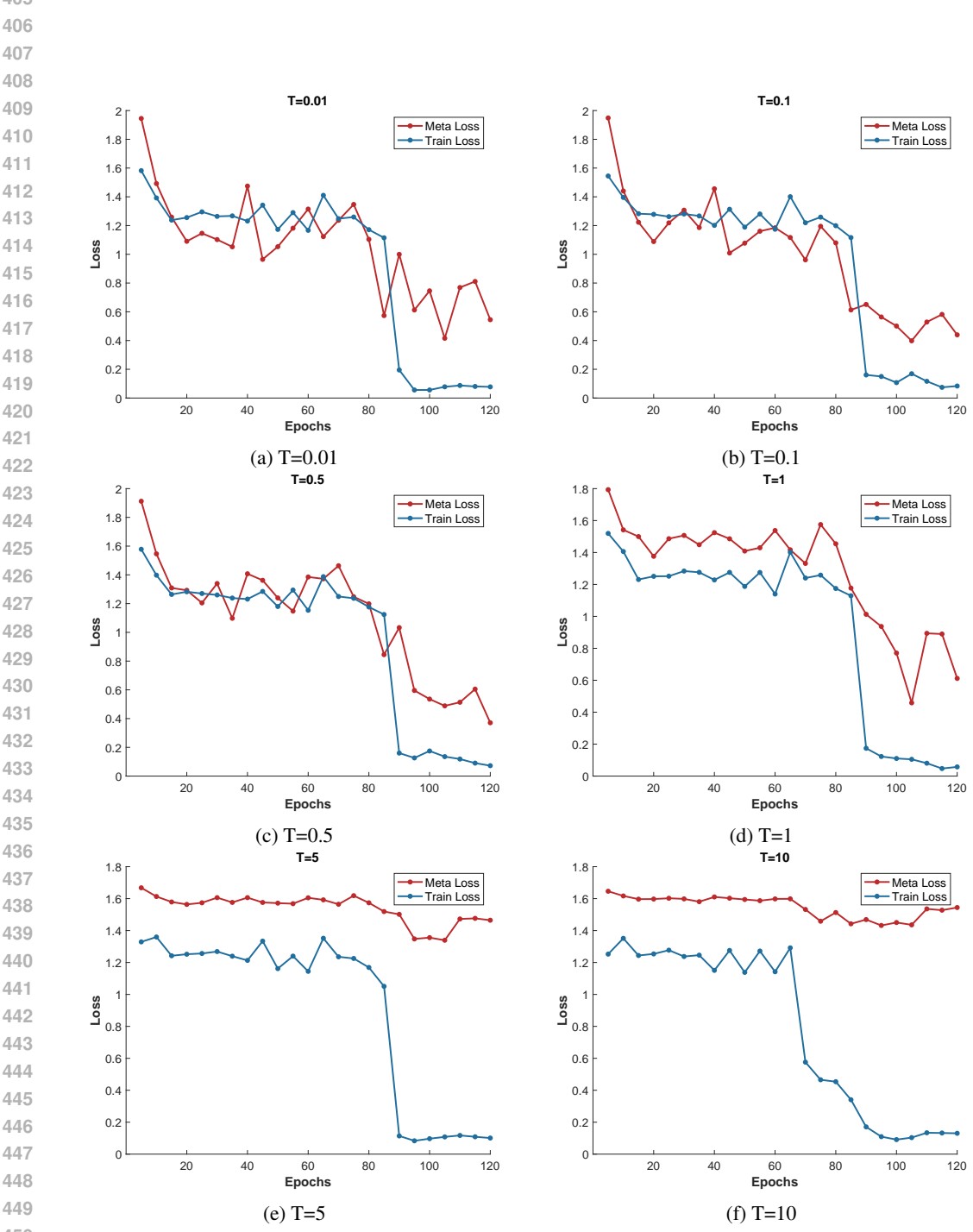

Figure 8: The convergence of Train Loss and Meta Loss for our proposed Conf-MWN method on CIFAR-10 datasets with 20% symmetric noise, showing results across different random trials under varying temperature parameters $T$.

### F.4 THE NUMBER OF CALIBRATION DATA

We have also conducted experiments on the CIFAR-10N datasets by varying the number of calibration data and the results are respectively summarized in Table 27. As can be observed from these results, a very small amount of data is sufficient for achieving a promising performance.

Table 27: `Ineff.` and `Cov.` results of Conf-MWN by varying the number of calibration data $n$ on CIFAR-10N datasets ("Worst" noise).

| $n$ | 5000 | 3000 | 1000 | 500 |
|---|---|---|---|---|
| Ineff. | 3.53 | 3.33 | 3.70 | 3.16 |
| Cov.(%) | 98.95 | 98.82 | 99.02 | 98.70 |

## G PREDICTION SET EXAMPLES UNDER CE AND CONF-MWN

This section provides examples of prediction sets by CE and Con-MWN on CIFAR-10 (Figure 9) and CIFAR-100 datasets (Figure 10) with training label noise.

## H THE USE OF LLMS

Large language models were used solely as a general-purpose tool for minor grammatical checks and assistance in generating visualization code. They played no role in the intellectual substance of this work.

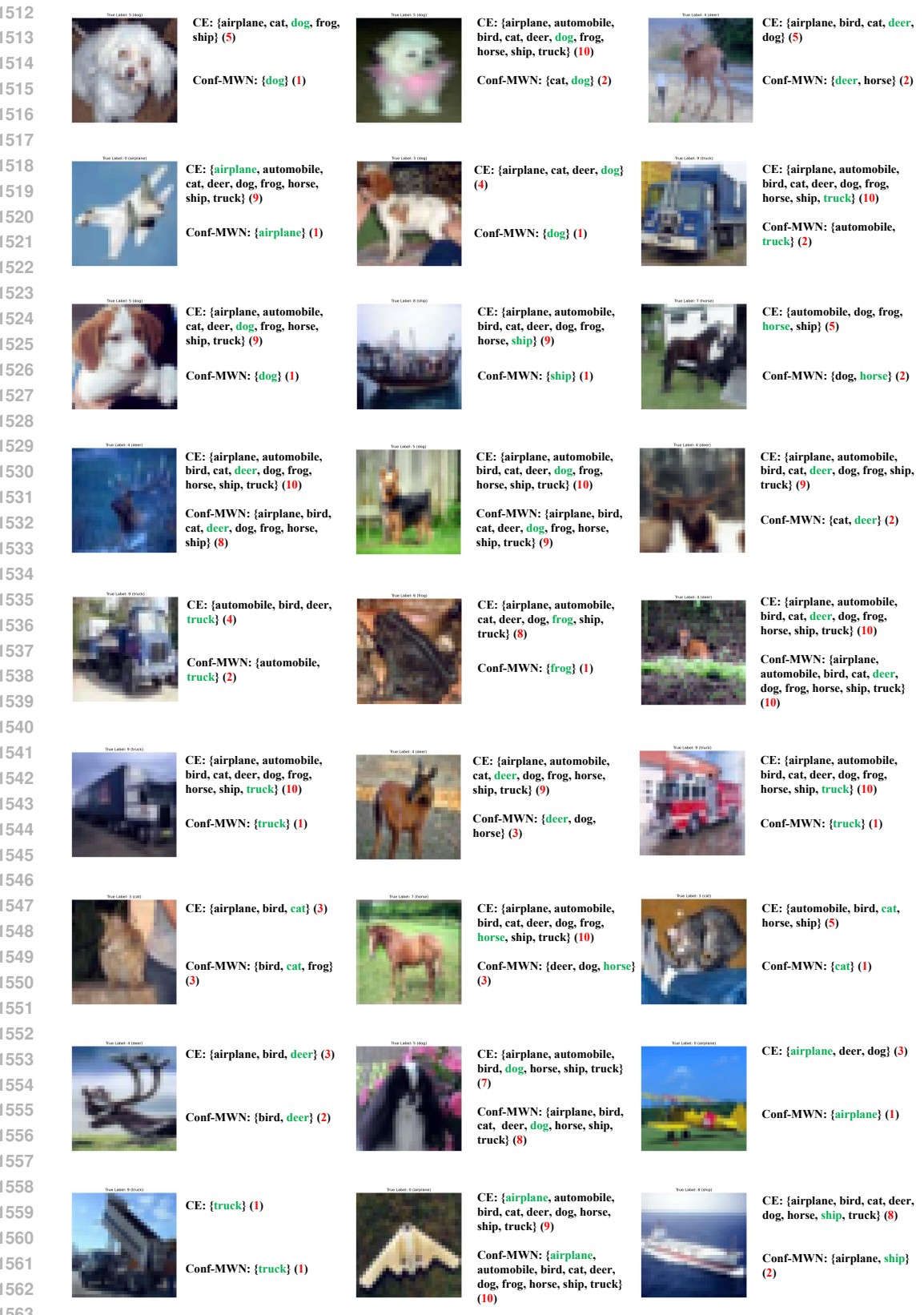

Figure 9: Examples of prediction sets obtained by CE and Conf-MWN (true labels are in green and the red numbers denote sizes). Models are trained on CIFAR-10 datasets with 40% symmetric noise.

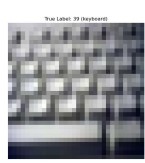

CE: {aquarium fish, baby, bear, bed, beetle, bowl, bus, castle, caterpillar, clock, cloud, couch, crocodile, fox, hamster, house, keyboard, leopard, lobster, maple tree, mouse, mushroom, oak tree, orange, pear, pickup truck, pine tree, plain, porcupine, raccoon, ray, seal, shark, shrew, skyscraper, snake, streetcar, tank, telephone, tiger, tulip, willow tree, worm} (43)

Conf-MWN: {keyboard} (1)

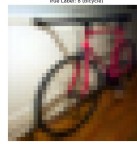

CE: {baby, beaver, bed, bee, beetle, bicycle, bottle, boy, bridge, butterfly, camel, can, cattle, chair, clock, cockroach, couch, crab, crocodile, cup, dinosaur, flatfish, forest, girl, kangaroo, lawn mower, lizard, lobster, man, motorcycle, mouse, orchid, otter, pear, plate, rabbit, rocket, rose, seal, shrew, snail, snake, spider, streetcar, table, tank, television, tiger, tractor, train, trout, tulip, wardrobe, whale} (54)

Conf-MWN: {apple, beetle, bicycle, bottle, can, clock, cockroach, cup, dinosaur, forest, lamp, lizard, lobster, motorcycle, mouse, mushroom, orchid, pear, plate, rabbit, rose, snail, spider, table, telephone, television, tiger, tractor, tulip, willow tree} (30)

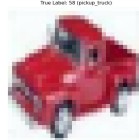

CE: {apple, aquarium fish, baby, beaver, bed, bee, beetle, bottle, bowl, boy, bus, can, caterpillar, hair, chimpanzee, clock, cockroach, crab, crocodile, dinosaur, dolphin, flatfish, forest, fox, girl, lamp, lawn mower, lizard, lobster, maple tree, motorcycle, mouse, oak tree, orchid, otter, pickup truck, plain, poppy, porcupine, possum, raccoon, rose, shark, shrew, snail, snake, sunflower, sweet pepper, table, tank, telephone, television, tiger, tractor, train, trout, turtle, wardrobe, whale, wolf, woman} (61)

Conf-MWN: {baby, bed, beetle, bowl, boy, bus, chair, cockroach, couch, crab, dinosaur, amp, lawn mower, lobster, maple tree, motorcycle, orchid, pickup truck, porcupine, rose, snake, sweet pepper, table, tank, telephone, tractor, trout} (27)

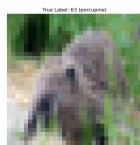

CE: {bear, beaver, castle, cattle, chimpanzee, crocodile, elephant, forest, kangaroo, leopard, lobster, mouse, oak tree, otter, plain, poppy, porcupine, possum, rabbit, raccoon, ray, road, sea, seal, shrew, skunk, squirrel, streetcar, sweet pepper, turtle, wolf} (31)

Conf-MWN: {bear, beaver, chimpanzee, crocodile, dinosaur, elephant, leopard, porcupine, possum, rabbit, raccoon, shrew, squirrel} (13)

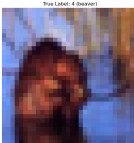

CE: {apple, baby, bear, beaver, bed, bee, beetle, bottle, boy, butterfly, camel, can, caterpillar, cattle, chimpanzee, clock, cloud, cockroach, couch, crab, elephant, flatfish, forest, girl, kangaroo, lamp, leopard, lion, lizard, lobster, man, maple tree, mountain, mouse, mushroom, otter, palm tree, pine tree, porcupine, possum, rabbit, raccoon ,ray, sea, seal, shrew, skunk, skyscraper, snail, snake, spider, squirrel, streetcar, sunflower, sweet pepper, table, telephone, television, tiger, tractor, train, trout, tulip, turtle, wardrobe, willow tree, woman} (67)

Conf-MWN: {bear, beaver, boy, butterfly, camel, girl, mouse, mushroom, otter, porcupine, possum, rabbit, ray, seal, shrew, skunk, snail, squirrel, turtle, woman} (20)

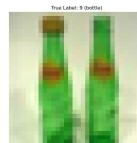

CE: {bottle, can, :caterpillar, lamp, pear, rocket, sunflower, sweet pepper} (8)

Conf-MWN: {bottle} (1)

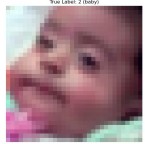

CE: {apple, baby, beaver, beetle, boy, cloud, forest, girl, hamster, lizard, lobster, man, mouse, oak tree, otter, pear, pickup truck, possum, ray, woman, worm} (21)

Conf-MWN: {baby, boy, girl, hamster} (27)

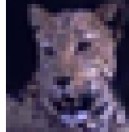

CE: {apple, baby, bear, beaver, bed, bee, beetle, bicycle, bottle, bowl, boy, bridge, bus, butterfly, can, castle, caterpillar, cattle, chair, chimpanzee, clock, cloud, cockroach, couch, crab, crocodile, cup, dinosaur, dolphin, elephant, forest, fox, girl, hamster, house, kangaroo, keyboard, lawn mower, leopard, lion, lizard, lobster, man, maple tree, motorcycle, mountain, mouse, mushroom, oak tree, orange, orchid, otter, pickup truck, pine tree, plain, plate, poppy, porcupine, possum, raccoon, ray, road, rocket, rose, sea, shark, shrew, skunk, snail, snake, spider, squirrel, streetcar, sunflower, sweet pepper, table, tank, telephone, television, tiger, tractor, train, tulip, turtle, wardrobe, willow tree, wolf, woman, worm} (89)

Conf-MWN: {leopard, tiger} (2)

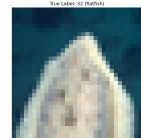

CE: {baby, bear, beaver, bicycle, bottle, boy, bridge, bus, butterfly, camel, castle, caterpillar, cattle, chimpanzee, clock, cloud, dinosaur, dolphin, elephant, flatfish, forest, girl, house, kangaroo, lamp, leopard, lion, lizard, man, mountain, mouse, oak tree, orange, otter, pear, pickup truck, pine tree, plate, poppy, porcupine, possum, raccoon, ray, road, rocket, sea, seal, shark, shrew, skunk, skyscraper, spider, streetcar, sunflower, sweet pepper, tank, telephone, television, tiger, train, tulip, turtle, : wardrobe, whale, willow tree, wolf, woman, worm} (68)

Conf-MWN: {camel, castle, flatfish, lamp, mountain, possum, ray, road, rocket, seal, shark, skyscraper, tulip, turtle, worm} (15)

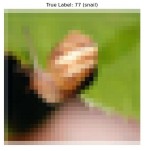

CE: {butterfly, clock, girl, rose, snail, spider, television, tulip} (8)

Conf-MWN: {beetle, butterfly, lizard, rose, snail, spider} (6)

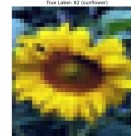

CE: {bear, bee, boy, bus, motorcycle, plain, poppy, rocket, shark, shrew, sunflower, tulip} (12)

Conf-MWN: {sunflower} (1)

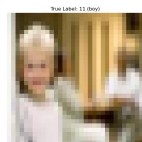

CE: {apple, aquarium fish, baby, bear, beetle, boy, castle, caterpillar, cockroach, couch, dinosaur, dolphin, elephant, flatfish, forest, fox, girl, hamster, keyboard, lion, lizard, lobster, man, mouse, mushroom, oak tree, orange, orchid, otter, palm tree, pear, plain, poppy, porcupine, possum, sea, skunk, skyscraper, snail, snake, spider, streetcar, table, television, tiger, tractor, train, tulip, turtle, willow tree, wolf, woman, worm} (53)

Conf-MWN: {apple, baby, boy, caterpillar, cockroach, girl, keyboard, lizard, man, mushroom, orange, pear, snail, squirrel, woman, worm} (16)

Figure 10: Examples of prediction sets obtained by CE and Conf-MWN (true labels are in green and the red numbers denote sizes). Models are trained on CIFAR-100 datasets with 20% symmetric noise.

