# OpenReview forum: "Efficient Conformal Prediction via Conformalized Meta-Learning from Noisy Labels"
_ICLR.cc/2026/Conference — Submitted to ICLR 2026_

### Official Review · Reviewer_pU7Y · 2025-10-15

**Soundness:** 2
**Presentation:** 2
**Contribution:** 2
**Rating:** 2
**Confidence:** 5

**Summary:**

The submission studies the impact of a model pre-trained with noisy labels on the performance of split CP. This is done by introducing some theoretical analysis of the average set size, as well as by proposing a meta-learning strategy in which clean-label data is used to enhance the efficiency of CP.

**Strengths:**

Understanding how the quality of a predictor affects the efficiency of CP is a valuable goal, and noisy labels offer a valid setting to model inaccuracies in the pre-trained predictor.

The submission builds on existing art to attempt an analysis of the role of noisy labels. Furthermore, the proposed meta-learning approach is sensible and it appears to be effective in practice.

**Weaknesses:**

The analysis builds directly on prior art, and it does not appear to add new substantial insights about the role of noisy labels. In particular, the only new result, Proposition 4.2 is formulated under strong, and not clearly expressed, assumptions, and it only provides an approximated result, namely (4). It is not clear what the approximate inequality (4) means in mathematical terms, and it is not clear what type of insights can be obtained from (4).

The proposed meta-learning strategy is quite straightforward. Furthermore, as also mentioned by the authors, there are many other approaches that could be attempted to operate on noisy-label data when reference clean-label data is present.

The experimental results are limited in scope, considering very few reference strategies.

**Questions:**

1) What is the significance of the assumptions underlying Proposition 4.2?

2) What does the approximation (4) mean at a mathematical level?

3) What type of insights can be obtained from (4)?

4) Which other methods could be considered that leverage clean-label data in addition to noisy-label data?

---

### Official Review · Reviewer_xkzW · 2025-10-29

**Soundness:** 3
**Presentation:** 3
**Contribution:** 2
**Rating:** 4
**Confidence:** 4

**Summary:**

The paper introduces a method that combines meta-learning with conformal training to enhance the efficiency of conformal classifiers trained on data with label noise. The approach jointly optimizes a classifier and a weight function in a bi-level optimization framework: the classifier minimizes cross-entropy loss, while the weight function minimizes prediction set size using a relaxed, differentiable conformal prediction procedure applied to a clean subset of samples. The proposed method demonstrates positive empirical results and is supported by theoretical analysis.

**Strengths:**

- The proposed method is conceptually simple yet empirically effective, as demonstrated by the reported results. This balance of simplicity and performance suggests that the approach could serve as a practical and valuable addition to the literature on conformal training, particularly in scenarios involving label noise.
- The paper is generally well-structured and easy to follow, even though it includes some uncommon terminology such as “deep learners” and minor typographical errors (e.g., “uncertainty qualification” instead of “uncertainty quantification”).
- The empirical evaluation is extensive and well documented in the appendix.

**Weaknesses:**

The primary weakness of the paper is that its contributions do not appear to offer substantial novelty or impact. The proposed approach combines existing techniques, namely meta-learning and conformal training, to improve conformal prediction under label noise. While the empirical results are encouraging, it remains unclear whether this combination represents a meaningful advancement beyond prior work.

For example, in Section 4, the empirical observation is not particularly surprising. It is well established that optimal (most efficient) prediction sets are obtained under the true distribution $P(Y|X)$ [1], and introducing label noise during training naturally pushes the model away from this distribution, resulting in larger prediction sets. Furthermore, the theoretical component does not clearly differentiate itself from previous results by Dhillon et al. (2024) and Zecchin et al. (2024). I recommend that the authors clarify these distinctions in the main text.

Regarding the meta-learning aspect, the contribution seems limited to leveraging a set of clean samples (without label noise) to learn a weight-function like proposed by Shu et al. (2019) by minimizing a relaxed conformal training loss like in Stutz et al. (2022). This appears to be a minor extension, and I invite the authors to clarify whether there is a deeper contribution that I may have overlooked. On the theoretical side, again, it is not evident how the main result advances beyond prior work.

There are a few minor presentation issues worth noting. In Theorem 4.1, it would improve readability to state the assumptions directly within the theorem rather than referring readers to the appendix. In Section 5, the parameters $\mathbf{w}$ and $\Theta$ should be introduced more clearly; I had to consult Shu et al. (2019) to fully understand their meaning.

### References

[1] Vovk, Vladimir, et al. "Criteria of efficiency for conformal prediction." Symposium on conformal and probabilistic prediction with applications. Cham: Springer International Publishing, 2016.

**Questions:**

Would it be possible to disentangle the effect of meta learning vs the conformal loss? From the experiments in Tables 1 and 2, it seems the conformal loss only marginally improves the results (making it worse in some cases), while meta learning is doing most of the heavy lifting. Meta learning improves coverage even in the absence of noise, which is interesting. Any intuition on why that could be? For instance, for CIFAR-100 with no label noise, meta learning improves coverage without improving accuracy, which suggests it improves the model calibration somehow.

---

### Official Review · Reviewer_aXem · 2025-10-30

**Soundness:** 3
**Presentation:** 3
**Contribution:** 3
**Rating:** 4
**Confidence:** 3

**Summary:**

This paper investigates the impact of label noise in training data on the efficiency (i.e., size) of prediction sets in conformal prediction for classification tasks. It provides empirical evidence and theoretical analysis demonstrating that noisy training labels lead to larger prediction sets while maintaining coverage guarantees, assuming a clean calibration set. To mitigate this, the authors propose Conf-MWN, an efficiency-aware meta-learning method that uses a small clean meta dataset to directly minimize the empirical prediction set size via sample re-weighting during classifier training. Experiments on synthetic and real noisy datasets show improved efficiency without sacrificing coverage.

**Strengths:**

The paper deals with a practical problem: the degradation of conformal prediction efficiency due to training label noise, distinct from prior focus on calibration noise. Theoretical analysis seems solid.

**Weaknesses:**

The novelty of the proposed method is somewhat limited as it is based of a combination of existing approaches for meta-learning and conformal training.
In addition, Propositions 4.1,4.2 rely on 0-1 score which is a bit non standard, and it is unclear how this score can be computed for the test set, where the true label is unknown.

**Questions:**

1.⁠ ⁠Inefficiency improvement over CE-MVN seems marginal for real datasets. Could this be justified?

2.⁠ ⁠Comparison to CE-MVN is missing in Tables 5-8. Consider adding this comparison to reassure the benefit of the proposed method.

3.⁠ ⁠Minor issues:
* ⁠Eq. (5) it seems that w is not explicitly defined.
* Line 161 - "can largely deviate" should be "deviate"

---

### Official Review · Reviewer_o2TS · 2025-10-31

**Soundness:** 3
**Presentation:** 3
**Contribution:** 3
**Rating:** 6
**Confidence:** 3

**Summary:**

This paper addresses the problem of prediction set inefficiency in conformal prediction (CP) when the base classifier is trained on noisy labels. The authors empirically show that label noise can significantly enlarge prediction sets while preserving coverage. To mitigate this, they propose an efficiency-aware Conformalized Meta-Weight-Net (Conf-MWN), which uses a small clean meta-dataset to reweight noisy training examples. The meta-objective directly minimizes the empirical prediction set size through a differentiable relaxation of the conformal quantile. Experiments show notable reductions in set size under both synthetic and real-world noise.

**Strengths:**

1. Addresses an underexplored yet practically relevant problem: label noise–induced inefficiency in conformal prediction.
2. The proposed Conf-MWN method is conceptually simple, computationally feasible, and empirically effective.
3. Experimental validation is broad and convincing, covering synthetic and real noisy datasets, multiple α-levels, and combinations with NACP.
4. Clarity and presentation quality are high; figures and tables are easy to follow.

**Weaknesses:**

1. Proofs apply only to trivial 0–1 nonconformity scores and do not support the HPS/APS-based experiments.
2. The method operates at the training stage while the inefficiency arises in calibration; no theoretical link is established between reweighting and quantile correction.
3. Assumption of a small clean meta-set is unrealistic for most high-noise domains; robustness to imperfect meta data is untested.
4. Replacing the CE meta-loss in MWN with a differentiable set-size surrogate is a moderate extension, not a conceptual leap.
5. Efficiency gains may simply result from accuracy improvements, not genuine uncertainty calibration.

**Questions:**

1. How sensitive is the method to the size and quality of the meta-dataset?
2. Can the authors provide accuracy-controlled comparisons or ablations to isolate the effect on inefficiency?
3. What happens if the meta-dataset also contains label noise?

---

### Meta-Review · Area_Chair_jPQT · 2026-01-09

**Summary:**

This paper studies the inefficiency of conformal prediction under training label noise and proposes an efficiency-aware conformalized meta-learning approach (Conf-MWN) that leverages a small clean meta-dataset to reduce prediction set size. Reviewers agreed that the problem is practically relevant and that the empirical results show consistent reductions in set size while preserving coverage. However, reviewers also raised concerns about the limited novelty and impact of the contribution, noting that the method largely combines existing ideas from meta-learning and conformal training. The theoretical analysis was found to be limited in scope and only partially aligned with the experimental setting. The AC recommends rejection.

**Reviewer Concerns:**

While the method is empirically effective, several key concerns remain. Theoretical results rely on restrictive assumptions and do not clearly support the main experimental settings using APS/HPS. The connection between training-stage reweighting and calibration-stage conformal quantiles is not theoretically established. Reviewers also questioned whether the observed efficiency gains stem from improved accuracy rather than improved uncertainty quantification. The authors did not provide a rebuttal.

**Reviewer Scores:**

Reviewer scores were mixed, but overall rejection.

---

### Decision · Program_Chairs · 2026-01-26

Reject